# A bacteriophage endolysin that eliminates intracellular streptococci

**Yang Shen[1†], Marilia Barros[2†], Tarek Vennemann[2], D Travis Gallagher[1,3], Yizhou Yin[1], Sara B Linden[1], Ryan D Heselpoth[1], Dennis J Spencer[4], David M Donovan[5], John Moult[1,6], Vincent A Fischetti[4], Frank Heinrich[2,7], Mathias Lösche[2,7,8], Daniel C Nelson[1,9]***

[1]Institute for Bioscience and Biotechnology Research, University of Maryland, College Park, Rockville, United States; [2]Department of Physics, Carnegie Mellon University, Pittsburgh, United States; [3]National Institute of Standards and Technology, Gaithersburg, United States; [4]Laboratory of Bacterial Pathogenesis and Immunology, The Rockefeller University, New York, United States; [5]Animal Biosciences and Biotechnology Lab, Agricultural Research Service, USDA, Beltsville, United States; [6]Department of Cell Biology and Molecular Genetics, University of Maryland, College Park, Rockville, United States; [7]Center for Neutron Research, National Institute of Standards and Technology, Gaithersburg, United States; [8]Department of Biomedical Engineering, Carnegie Mellon University, Pittsburgh, United States; [9]Department of Veterinary Medicine, University of Maryland, College Park, College Park, United States

**\*For correspondence:** nelsond@umd.edu

[†]These authors contributed equally to this work

**Abstract** PlyC, a bacteriophage-encoded endolysin, lyses *Streptococcus pyogenes* (*Spy*) on contact. Here, we demonstrate that PlyC is a potent agent for controlling intracellular *Spy* that often underlies refractory infections. We show that the PlyC holoenzyme, mediated by its PlyCB subunit, crosses epithelial cell membranes and clears intracellular *Spy* in a dose-dependent manner. Quantitative studies using model membranes establish that PlyCB interacts strongly with phosphatidylserine (PS), whereas its interaction with other lipids is weak, suggesting specificity for PS as its cellular receptor. Neutron reflection further substantiates that PlyC penetrates bilayers above a PS threshold concentration. Crystallography and docking studies identify key residues that mediate PlyCB–PS interactions, which are validated by site-directed mutagenesis. This is the first report that a native endolysin can traverse epithelial membranes, thus substantiating the potential of PlyC as an antimicrobial for *Spy* in the extracellular and intracellular milieu and as a scaffold for engineering other functionalities.

## Introduction

*Streptococcus pyogenes (Spy)*, a leading human pathogen, colonizes the skin and mucosal surface, and can lead to severe life-threatening conditions including bacteremia, necrotizing fasciitis, and streptococcal toxic shock syndrome (*Cunningham, 2000*), as well as a variety of superficial infections such as impetigo and pharyngitis. Streptococcal pharyngitis, one of the most common childhood illnesses in the United States, accounts for 15 million outpatient physician visits (*Hing et al., 2008*) and leads to an estimated economic burden of more than half a billion dollar per year (*Pfoh et al., 2008*). Globally, *Spy*-associated diseases are correlated with high mortality and represent a worldwide health concern (*Carapetis et al., 2005*).

**eLife digest** *Streptococcus pyogenes* is the bacterium that causes throat infections and other serious infections in humans. Antibiotics such as penicillin are used to treat active infections, but so-called "strep throat infections" often return after treatment. This is because *S. pyogenes* can enter the cells that line the throat and hide from the antibiotics, which cannot enter the throat cells.

Endolysins are enzymes produced by viruses that attack bacteria, and these enzymes target and destroy the bacterial cell wall. A previous study revealed that an endolysin known as PlyC could destroy *S. pyogenes* bacteria on contact. PlyC and other endolysins have the potential to act as alternatives to common antibiotics, but before these enzymes can be developed as therapeutics, it is important to understand how they interact with human host cells.

Like antibiotics, the PlyC endolysin was not expected to enter throat cells. However, Shen, Barros et al. have now discovered that not only can PlyC enter throat cells, it can essentially chase down and kill *S. pyogenes* that are hiding inside. Other similar enzymes could not act in this way, and further studies confirmed that PlyC could move around inside a throat cell without causing it damage. Shen, Barros et al. also determined that PlyC has a pocket on its surface that binds with a specific component of the throat cell membrane, a molecule called phosphatidylserine. This interaction – which is a bit like a lock and key – grants PlyC access into the cell.

While it is clear that PlyC eventually kills *S. pyogenes* hiding inside throat cells, future experiments will aim to determine how PlyC moves around once inside an infected throat cell. Together, an understanding of how an endolysin enters cells and destroys hiding *S. pyogenes* will contribute to the development of endolysins with broader activity, which can be used as alternatives to common antibiotics.

The ability of many pathogens, such as *Spy*, to invade host cells, survive at low levels in an intra-cellular niche, and seed reinfection poses a major challenge in developing effective treatment proto-cols to address this carrier state (*Rohde and Cleary, 2016*). Inside host cells, pathogens are protected from many antimicrobial agents and host immune effectors, and these evasion mecha-nisms directly contribute to long-term persistence (*Helaine et al., 2014*; *Iovino et al., 2014*; *Pentecost et al., 2010*). *Spy* is well known for its ability to proliferate within host cells (*Barnett et al., 2013*) and escape autophagic degradation (*Sakurai et al., 2010*). Notably, *Spy* can be recovered from clinical specimens of excised human tonsils (*Osterlund et al., 1997*), even after antibiotic treatment. No effective approach has yet been identified that can specifically kill intracellu-lar *Spy*, although successful strategies have been reported to control other intracellular bacteria, for example, by antibiotics that can accumulate within cells (*Ahren et al., 2002*; *Mandell and Coleman, 2000*), encapsulating antibiotics into liposomes or nanoparticles for intracellular delivery (*Couvreur et al., 1991*), or antimicrobial peptides (*Di Grazia et al., 2014*). There is thus the need for an antimicrobial agent that targets both extracellular *and* intracellular *Spy*.

Bacteriophage-encoded endolysins that act as peptidoglycan hydrolases represent a novel class of antimicrobials that kill bacteria by lysing the cell wall. Exogenously applied, these enzymes kill sus-ceptible Gram-positive bacteria rapidly and with high specificity (*Loessner, 2005*). Numerous studies demonstrate the therapeutic potential of endolysins in various animal infection models of human dis-ease, and no adverse side effects have yet been reported (*Daniel et al., 2010*; *Pastagia et al., 2013*; *Schuch et al., 2014*). Therefore, endolysins have recently gained increasing attention as anti-microbial agents or 'enzybiotics' (*Fischetti, 2005*). However, despite a wealth of knowledge about their antibacterial potential, little is known about their capacity, either inherent or conferred by molecular engineering, to penetrate mammalian cells where they could target intracellular pathogens.

One such *Spy*-specific endolysin, PlyC, has been characterized using biochemical and structural techniques and shown to be a 114 kDa multimeric holoenzyme. These studies demonstrated that PlyC comprises one PlyCA subunit, harboring two distinct catalytic domains that work synergistically to cleave two different bonds in the peptidoglycan, and eight identical PlyCB subunits that form a symmetrical ring for cell wall recognition (*McGowan et al., 2012*; *Nelson et al., 2006*). PlyC's

bactericidal activity against *Spy* in biofilms was shown in vitro (*Shen et al., 2013*), as was its therapeutic potential in an in vivo model of upper respiratory *Spy* colonization (*Nelson et al., 2001*). Here, we investigate the ability of PlyC to target and control intracellular *Spy* believed to be associated with streptococcal infections that are highly refractory to antibiotic treatment.

## Results

### PlyC possesses an inherent activity against intracellular *Spy*

To mimic the pathogenic niche for *Spy* colonization and invasion, we established a co-culture model of human epithelial cells and *Spy* strain D471 to differentiate non-adherent, adherent, and intracellular streptococci. In experiments with human epithelial cell lines A549 (*Figure 1—figure supplement 1*) or Detroit 562 (data not shown), rates of *Spy* adherence ranged from 1 to 5% of the inoculum and rates of internalization ranged from 1 to 10% of the adherent streptococci, which are consistent with previous *Spy*/epithelial cell co-culture studies (*LaPenta et al., 1994*; *Ryan et al., 2001*) as well as in vivo findings (*Kugelberg et al., 2005*). Using this co-culture system, the bacteriolytic efficacy of three endolysins known to lyse *Spy* in vitro (PlyC (*Nelson et al., 2006*); B30 (*Donovan et al., 2006*); and Ply700 (*Celia et al., 2008*)) were evaluated for activity against intracellular *Spy*. Although we anticipated the enzymes would need to be engineered with cell penetrating peptides or transduction domains in order to acheive intracellular activity, treatment with 50 µg/ml WT PlyC reduced intracellular colonization (colony forming units; CFUs) by 95% (p<0.001) within 1 hr, and lower concentrations displayed a dose response (*Figure 1a*). In contrast, the B30 and Ply700 streptococcal phage endolysins failed to significantly decrease the intracellular *Spy* CFUs. We then assessed PlyC in a co-culture with primary human tonsillar epithelial cells grown from a tonsillectomy as a more clinically relevant model since these cells are known to be the major reservoir for recurrent *Spy* pharyngotonsillitis (*Osterlund et al., 1997*). Roughly 90% of intracellular *Spy* were eliminated when treated with 50 µg/ml PlyC (*Figure 1b*), similar to the effect in immortalized A549 epithelial cells, although the lower dose treatments did not demonstrate significant killing. At present, it is not known if the differences in efficacy are due to differences in the distribution of cellular receptors between the cell types or other phenotypic differences. Nonetheless, these data indicate that the native PlyC holoenzyme can be internalized by mammalian cells and that the endolysin retains bacteriolytic efficacy against *Spy* in the intracellular environment.

### PlyCB mediates PlyC internalization, leading to colocalization with *Spy* and intracellular lysis of streptococci

One explanation for the demonstrated intracellular lytic activity of PlyC is that this finding could be an artifact if the enzyme damages plasma membranes or is otherwise toxic toward eukaryotic cells. To test for this possibility, we used propidium iodide (PI) to visually distinguish between cells with and without membrane damage. Thirty-minute exposures to 100 µg/ml PlyC revealed no membrane damage compared to Triton X-100 treated controls (*Figure 2a*). In addition, a trypan blue dye exclusion assay showed no difference between growth media control and 1 hr treatment with 100 µg/ml PlyC (data not shown). Both assays suggest that exposure to or internalization of PlyC does not compromise the membrane integrity of epithelial cells.

Next, we determined whether internalization requires the PlyC holoenzyme or is specific to one of the PlyC subunits. PlyC, PlyCA, and PlyCB were purified (*Figure 2—figure supplement 1*) and fluorescently labeled. At a concentration of 5 µg/ml, both the fluorescently PlyC holoenzyme and the PlyCB octamer, but not the PlyCA subunit, were internalized by A549 epithelial cells within 30 min (*Figure 2b*). Both internalized PlyC and PlyCB were found to accumulate in vesicle-like structures with diameters of ≈0.5 µm. In confocal microscopy experiments, A549 epithelial cells were exposed to *Spy* and 30 min later to PlyC. Wild type (WT) PlyC, with full bacteriolytic activity, colocalized with intracellular *Spy* (*Figure 2c*, merged image in the lower middle panel), where ruptured *Spy* cell walls are visible. No intact chain-like streptococci were observed through the entire Z-stack. Indeed, the maximum intensity projection (i.e. merged Z-stack) shows that all intracellular *Spy* were targeted and ruptured by internalized PlyC (*Figure 2c*, lower right), and no extracellular *Spy* could be detected outside or between the host cell boundaries (indicated by actin filament staining). Taken together, the data show that the subunit PlyCB mediates translocation of the PlyC

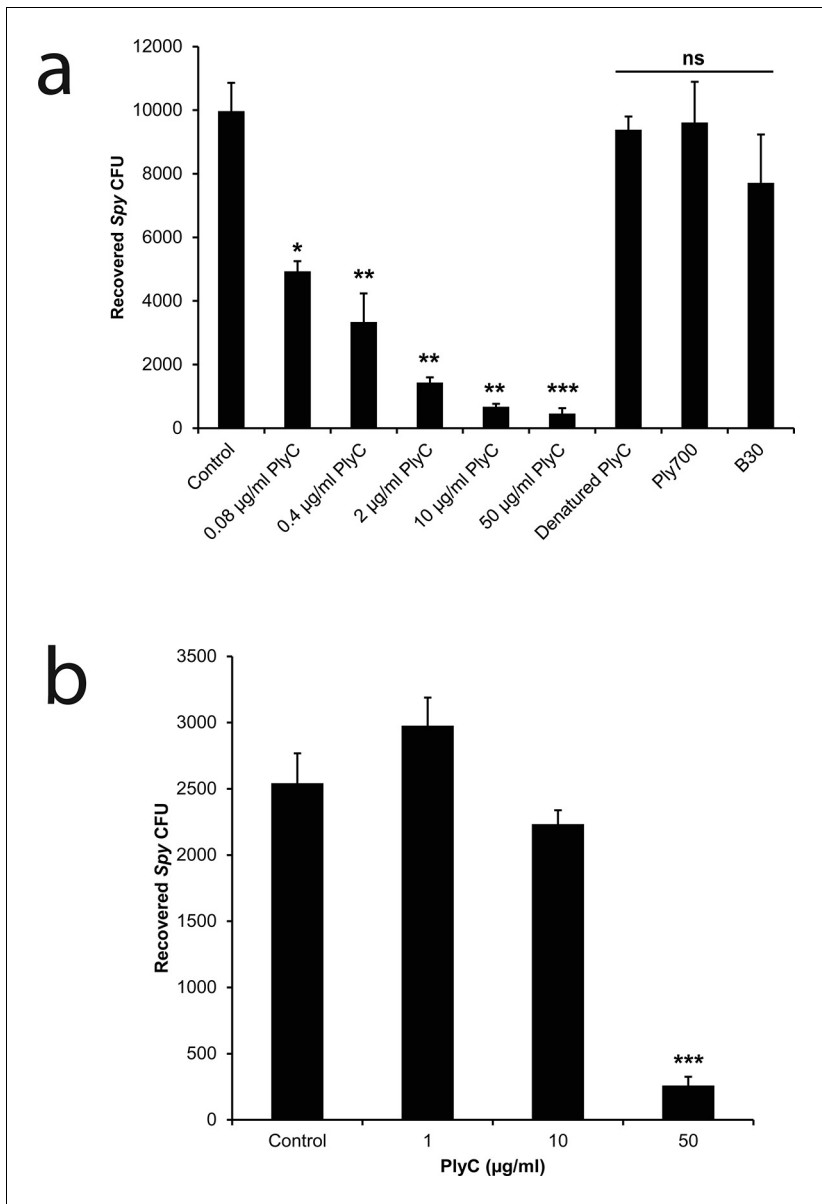

**Figure 1.** PlyC eliminates intracellular *Spy* in a dose-dependent manner. (**a**) *Spy*/A549 co-cultures, treated with 10 µg/ml penicillin and 200 µg/ml gentamicin for 1 hr to remove extracellular bacteria, were incubated for another hour with antibiotic-free growth medium in dilutions of PlyC beginning at 50 µg/ml (440 nM), or with 50 µg/ml heat-denaturated PlyC (70°C for 30 min.), 50 µg/ml (1890 nM) Ply700, or 50 µg/ml (890 nM) B30. Viable intracellular *Spy* were enumerated as colonies on agar plates that had been incubated with serial dilutions of cell lysates. (**b**) A primary tonsillar epithelial cell co-culture was treated with 10 µg/ml penicillin and 200 µg/ml gentamicin for 1 hr after which the antibiotic was removed and the tonsil cells were incubated with 50, 10, or 1 µg/ml PlyC for 1h before lysis and enumeration of *Spy*. All experiments were performed in triplicate biological replicates, with mean and standard deviations displayed. Statistical analysis using Student's t-test is reported as *p<0.05; **p<0.005; ***p<0.001; ns: not significant.

The following figure supplement is available for figure 1:

**Figure supplement 1.** Adherent and internalized colony counts of *Spy* recovered from co-culture assay.

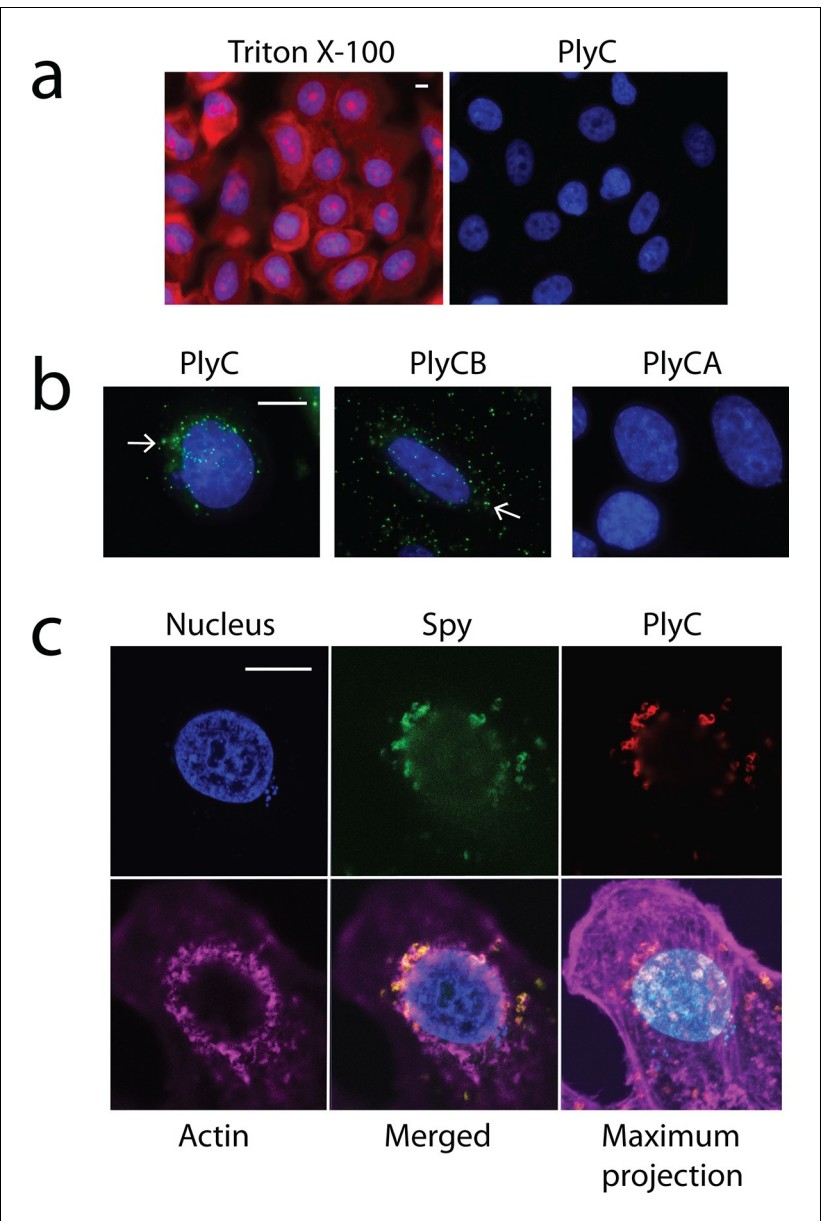

**Figure 2.** Membrane integrity is not compromised by PlyC, and internalization is mediated by the PlyCB subunit. (a) A549 epithelial cells were permeabilized with 0.02% Triton X-100 (left panel), as shown by staining with propidium iodide (red) and DAPI (nucleus, blue), whereas cells pre-incubated with 100 μg/ml PlyC for 1 hr showed no damage (right). Scale bar is 5 μm. (b) Fluorescently labeled PlyC (left panel) and PlyCB (middle panel), but not PlyCA (right panel), are internalized in A549 cells upon incubation, as shown with protein (5 μg/ml) conjugated with AlexaFluor555 (green stain). Cells were incubated in serum-free F12K medium for 30 min at 37°C, fixed with 4% PFA in PBS and subsequently stained with DAPI (blue). Arrows indicate internalized PlyC or PlyCB in vesicle-like structures with an average size of 0.5 μm. Scale bar is 10 μm. (c) Confocal microscopy of internalized PlyC colocalized with intracellular *Spy*. Nucleus (blue, DAPI), *Spy* bacteria (green, AlexFluor 488 conjugated wheat germ agglutinin), PlyC (red, AlexFluor 555 conjugated), and actin filament (magenta, AlexFluor 647 conjugated phalloidin) are shown in the same focal plane. PlyC (red) is colocalized with *Spy* (green) in the merged image. The maximum intensity projection shows all Z-stacks simultaneously. Scale bar, 10 μm.

The following figure supplement is available for figure 2:

**Figure supplement 1.** Purification of PlyC and its individual subunits.

holoenzyme across the plasma membrane, where it is subsequently trafficked and quantitatively lyses intracellular *Spy*.

## Key cationic residues of PlyCB play roles in *Spy* lytic activity, *Spy* binding, and epithelial cell internalization

The PlyC crystal structure (*McGowan et al., 2012*) provided key insights into the origin of PlyCB–membrane interactions. At neutral pH, the PlyCB octamer is net positive (pI = 8.5), and its solvent accessible surface contains 48 cationic residues, two arginines and four lysines on each monomer (*Figure 3a*). Cationic amino acid patches are known to frequently promote membrane adhesion and often result in membrane penetration of proteins, as described for cell-penetrating peptides (CPPs) (*Dietz, 2010*). To explore the roles of cationic residues in PlyC, we mutated the exposed Arg and Lys residues on PlyCB to Glu in order to independently determine the role of each mutation in *Spy* binding and lysis, as well as PlyC cellular uptake. Four of five holoenzyme constructs, PlyC(PlyCB$_{K70E, K71E}$), PlyC(PlyCB$_{K23E}$), PlyC(PlyCB$_{K59E}$), and PlyC(PlyCB$_{R66E}$), were soluble and tested for lytic activity in a turbidity reduction assay. A fifth mutant, PlyC(PlyCB$_{R29E}$) was insoluble and could not be further studied. The holoenzymes incorporating PlyCB$_{K23E}$ and PlyCB$_{K70E,K71E}$ were fully active in external *Spy* degradation, but PlyC(PlyCB$_{K59E}$) and PlyC(PlyCB$_{R66E}$) only had ≈ 15% and 1% of WT PlyC activity, respectively (*Figure 3b*).

In its role in external *Spy* degradation, PlyCB is known to bind the streptococcal surface (*Nelson et al., 2006*). Therefore, the same cationic to anionic mutations were made to PlyCB in the absence of the catalytic PlyCA and proper secondary structure folding was confirmed by infrared spectroscopy (*Figure 3—figure supplement 1*) and analytical gel filtration (data not shown). Of the four PlyCB constructs, only PlyCB$_{K70E,K71E}$ was insoluble and excluded from further study. The remaining PlyCB mutants were used to assess *Spy* binding and internalization by A549 epithelial cells. Fluorescence microscopy showed a lack of binding of the PlyCB$_{R66E}$ mutant to *Spy* while PlyCB$_{K23E}$ and PlyCB$_{K59E}$, as well as a charge-conserving PlyCB$_{R66K}$ mutant bound comparably to WT (*Figure 3c*, left). Moreover, the PlyCB$_{K23E}$, PlyCB$_{K59E}$, and PlyCB$_{R66E}$ mutants, but not PlyCB$_{R66K}$, lost the ability to enter A549 epithelial cells (*Figure 3c*, right). Thus, K23 of PlyCB is not involved in *Spy* binding but is involved in epithelial cell internalization, whereas K59 and, to a greater extent, R66 are involved in both *Spy* binding and internalization. In addition, the response of the charge-conserving PlyCB$_{R66K}$ mutant supports the hypothesis that electrostatic interaction involving R66 plays a significant role in PlyCB targeting both the *Spy* cell wall and the epithelial surface.

## PlyCB interacts specifically with anionic phospholipids

To identify ligands on the plasma membrane that mediate PlyCB internalization, two approaches were taken. First, we investigated whether electrostatic interactions with glycosaminoglycans (GAGs) play a role in PlyCB internalization, as has been shown for cationic CPPs (*Tyagi et al., 2001*) and can be assessed by competing for the membrane-bound GAGs with soluble GAGs such as heparin or chondroitin sulfate-B (CS-B). An excess of heparin or CS-B did not block PlyCB internalization into A549 epithelial cells (*Figure 4—figure supplement 1a–c*). Furthermore, treatment of A549 cells with chondroitinase ABC or heparinase III prior to PlyCB incubation did not abrogate PlyCB internalization (*Figure 4—figure supplement 1d,e*), ruling out a role for these GAGs as a PlyCB receptor.

The second approach focused on anionic lipids, which are also implicated in the internalization of cationic CPPs (*Su et al., 2010*). A phosphoinositide array (PIP Strip) showed that WT PlyCB interacts with phosphatidylinositol (PI), phosphatidic acid (PA), and phosphatidylserine (PS) but not with phosphatidylethanolamine (PE), phosphatidylcholine (PC), or any of the phosphatidylinositides (*Figure 4—figure supplement 2*), suggesting that the binding has elements of specificity. PlyCB$_{K59E}$ and PlyCB$_{R66E}$ that lack the ability to internalize in epithelial cells also lost the ability to bind PI and showed diminished binding to PA and PS.

Surface plasmon resonance (SPR) was subsequently used to quantify PlyCB binding to sparsely-tethered bilayer membranes (stBLMs) (*McGillivray et al., 2007*) containing various anionic lipids in a background of 1,2-dioleoyl-sn-glycero-3-phosphocholine (DOPC), where control experiments on pure DOPC showed no detectable protein binding. Typical SPR traces (*Figure 4a,b*) show that PlyCB binds phospholipids with largely different affinities and is presumably differentially organized at the surface of DOPC membranes that contain below and above 15 mol% 1,2-dioleoyl-sn-glycero-3-

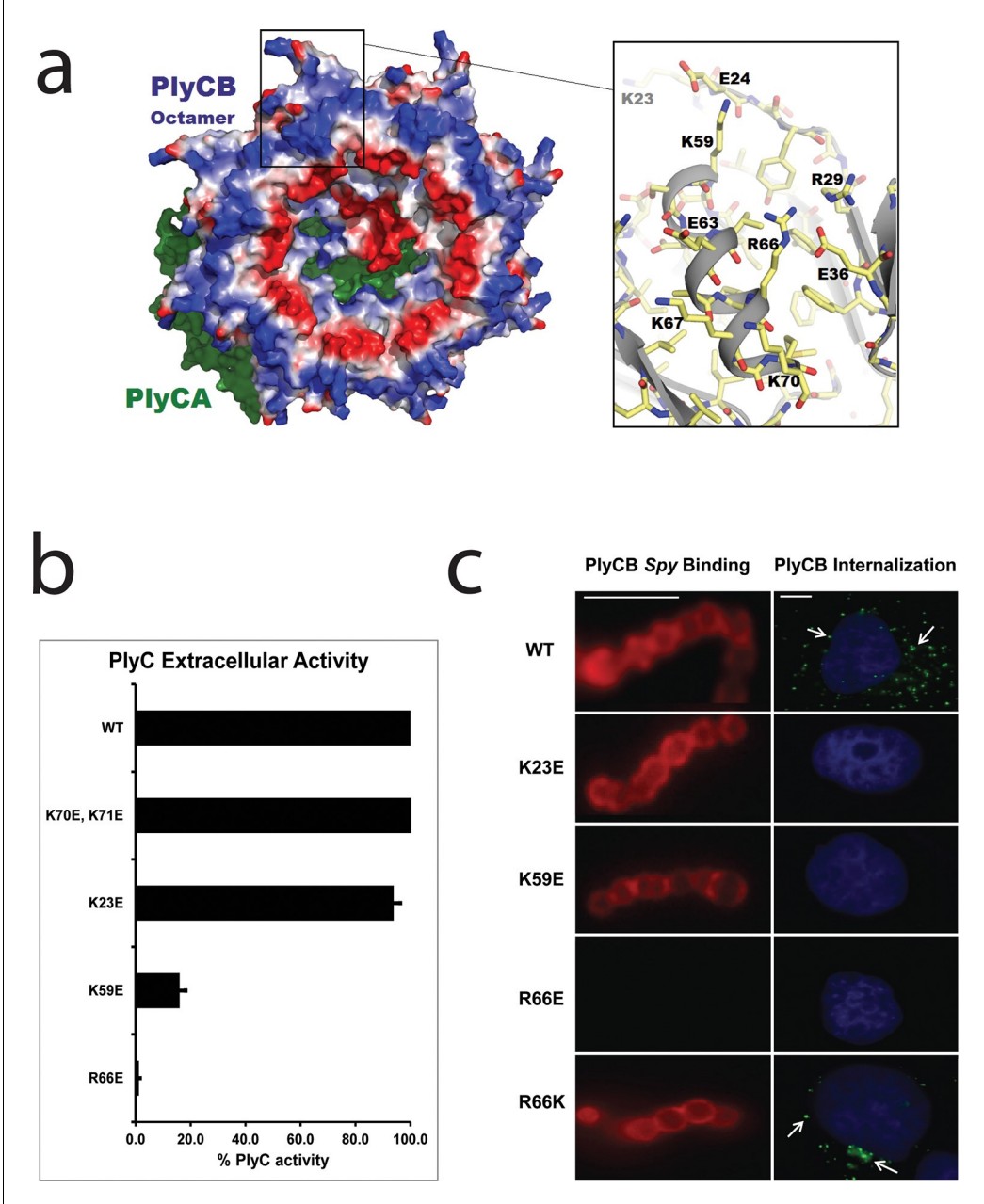

**Figure 3.** Cationic residues on the surface of PlyCB are important for *Spy* binding and epithelial cell internalization. (a) View of the membrane-binding interface of the PlyC holoenzyme (PDB: 4F88). The PlyCB octamer's solvent-accessible surface is color-coded blue and red for areas of net positive and negative charge, respectively, with Arg and Lys side chains assigned as positive and Asp and Glu as negative. The PlyCA catalytic subunit, located behind the PlyCB octamer in this view, is shown in green. The rectangular region at top is enlarged at the right using the high-resolution model of PlyCB (PDB: 4F87) that shows details of a cationic groove. Based on effects of mutations, this region appears important for interactions with both bacterial and mammalian cells. (b) Extracellular bacteriolytic activity of selected PlyC mutants on *Spy*, normalized to WT PlyC in a turbidity reduction assay. All experiments were performed in triplicate biological replicates, with mean and standard deviations displayed. (c) Microscopy of fluorescently labeled PlyCB and mutants on *Spy* (left column) and internalization into A549 epithelial cells (right column), where arrows point to intracellular PlyCB. In distinction to the data in *Figure 2c*, *Spy* are not ruptured upon protein binding since the constructs lack the catalytic PlyCA domains. Scale bar is 5 μm in the left column and 10 μm in the right column.

The following figure supplement is available for figure 3:

*Figure 3 continued on next page*

*Figure 3 continued*

**Figure supplement 1.** Fourier Transform Infrared Spectroscopy of WT PlyCB and binding groove mutants.

phospho-L-serine (DOPS). At 10 mol% PS, adsorbed PlyCB washes out in a buffer rinse whereas it is largely retained upon rinsing in membranes with 20 mol% PS. Simultaneously with the SPR characterization, electrochemical impedance spectroscopy (EIS) (*McGillivray et al., 2007*) was used to monitor bilayer degradation upon protein binding. The Cole-Cole plot in *Figure 4c*, obtained with 30 mol% PS at the highest PlyCB concentration, illustrates that changes in electrical membrane characteristics were minimal. The EIS spectra were measured between 100 kHz and 1 Hz, but the spectra shown here cover only 100 kHz to 13.6 Hz in order to visualize the capacitive semi-circles well. Neither the membrane capacitance $C_{mem}$ nor the membrane resistance $R_{def}$ (*Figure 4c*, inset) changed significantly (for quantitative assessment, see *Table 1*).

Raw SPR data, such as those in *Figure 4a,b*, were modeled in terms of Hill functions to determine protein affinities, $K_d$, to the membranes. *Figure 5a* displays results obtained with 30 mol% of various anionic lipids in DOPC-based stBLMs. PlyCB binds poorly to phosphatidylglycerol (PG) and PI and shows only low affinity to PA. In all these cases, rinsing the membranes with protein-free buffer reverts the SPR signal to baseline. In contrast, PlyCB binding to 30 mol% PS is extremely strong. *Figure 5b* shows PlyCB binding to stBLMs with PS concentrations between 10 and 30 mol%. At PS $\geq$20 mol%, the adsorption isotherms no longer followed a Hill function with $N = 1$ (i.e., a simple Langmuir isotherm) but were well described by Hill functions with $N > 1$ (*Table 2*). On membranes that contained 30 mol% PS, both the affinity of the protein ($K_d = 40 \pm 10$ nM) and its surface density were very high. Buffer rinses of stBLMs that contained PS at $\geq$15 mol% removed the bound protein only partially (*Figure 4b*). While membrane binding of PlyCB is extremely strong at high PS concentration, its affinity decreases rapidly at lower PS content. At 15 mol% PS, the $K_d$ increases to $\approx$0.5 µM, and at 10 mol% PS, PlyCB binding is reduced to the level of binding equal to 30 mol% PI or PG. This suggests that there is a threshold of PlyCB binding to PS-containing bilayers, where membranes at PS concentrations >15 mol% behave differently from membranes at $\leq$15 mol%.

The observed threshold in PlyCB binding to PS-containing membranes suggests that both electrostatic attraction and specific binding contribute to PlyCB interacting with the membrane. If this is true, anionic lipids distinct from PS can provide the electrostatic component of the interaction. We measured the affinity of the protein to an stBLM composed of PC:PA:PS = 70:20:10 (*Figure 5c*) and observed that the affinity was higher ($K_d \approx 1.2$ µM) than to membranes with 10 mol% of PS or 30 mol% PA as the only anionic lipid, but not as strong as to stBLMs containing $\geq$20 mol% PS. To confirm the PS specificity of PlyCB interaction with the membrane, we also measured the membrane affinity of PlyCB$_{R66E}$ (*Figure 5d*). As expected, this mutant binds poorly to PS, comparable with WT PlyCB to PI-containing membranes (compare *Figure 5d* and *Figure 5a*). These $K_d$ values were converted into the free energy, $\Delta G = -k_B T \ln K_d$, between the membrane-bound and dissolved states of the protein (details, see 'Materials and methods'). The major results of the series of experiments reported here are summarized in *Figure 6* in terms of $\Delta\Delta G$ values, that is, changes in $\Delta G$ compared with that of DOPC with 30 mol% DOPS, the membrane we found most strongly attracted the protein and therefore had a strongly negative $\Delta G$.

## Molecular details of PlyCB association with PS-containing membranes investigated with neutron reflection

Neutron reflection (NR) measurements prior to protein incubation confirmed that stBLMs of various anionic compositions all contained high-quality lipid membranes. Subsequently, neutron data were collected at multiple protein concentrations and, eventually, after a final rinse with protein-free buffer. In order to obtain high signal-to-noise data, PlyCB concentrations were chosen for which SPR showed high protein loading at the membrane surface. Each of the measurements was performed at multiple isotopic water (H$_2$O or D$_2$O) compositions of the buffer. This contrast-variation approach permits the determination of unique bilayer-protein complex structures, as shown earlier (*McGillivray et al., 2009*; *Shenoy et al. 2012b*), in terms of one-dimensional lipid/protein distributions as a function of the distance from the interface, *z*. These distributions are referred to as component volume occupancy (CVO) profiles (*Heinrich and Lösche, 2014*).

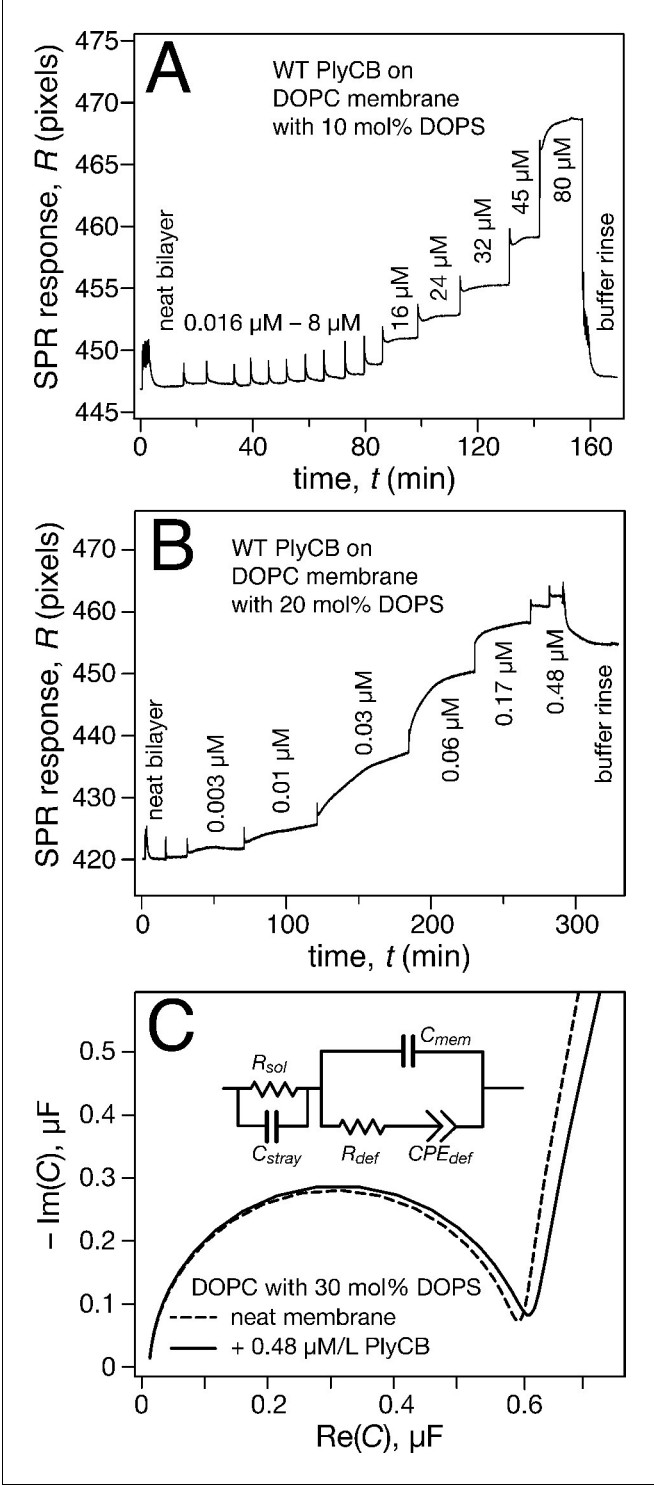

**Figure 4.** PlyCB binds to PS-containing membranes. PlyCB in 150 mM NaCl at pH 7.4 was characterized on surface-ligated DOPC-based stBLMs with PS at various concentrations and its impact on membrane structure monitored with EIS. (a,b) Binding of PlyCB to stBLMs with 10 and 20 mol% DOPS, respectively. Each spike indicates an injection with an increased concentration of protein. A final buffer rinse removed the adsorbed protein quantitatively on 10 mol% PS, but only partially on 20 mol% PS, indicating that the association of the protein with the membrane surface differed depending on PS concentration. (c) Raw EIS spectra (Cole-Cole plots) of an stBLM (20 mol% DOPS) as prepared (dashed line) and after incubation with 480 nM PlyCB (solid line). Fitted parameters (equivalent circuit model, see inset), corrected for electrode surface size, are given in *Table 1*.

*Figure 4 continued on next page*

*Figure 4 continued*

The following figure supplements are available for figure 4:

**Figure supplement 1.** Effects of soluble GAGs or GAG lyases on internalization of PlyC.

**Figure supplement 2.** Assessment of PlyCB binding to phospholipids using a PIP Strip array.

PlyCB-membrane complexes on PC/PS stBLMs organize in different overall structures on 15 and 30 mol% DOPS, as shown in *Figure 7*. At 15 mol% PS, PlyCB forms a diffuse layer of protein that extends about 200 Å out from the bilayer surface (*Figure 7a*). Overall, the lateral protein density remains low at any z. Moreover, protein insertion into the membrane is negligible. The CVO profiles at 15 mol% PS in the bilayer suggest that PlyCB forms an open-mesh network on the membrane surface that has a thickness of ≈ 20 nm, and does not penetrate the bilayer.

At high DOPS concentration, PlyCB associates with stBLMs in a different complex. *Figure 7b* shows the CVO profile of the protein on DOPC:DOPS 70:30 at $c_p$ = 600 nM. The protein is deeply inserted in the membrane and spans its entire thickness, headgroup to headgroup. Some protein is superficially adsorbed as evidenced by a protein CVO peak at a distance of ≈ 20 Å above the membrane surface. Flushing the membrane with protein-free buffer removed this adsorbed protein, as shown in *Figure 7c*. The CVO profile after the flush shows that the PlyCB protein retained within the membrane is almost symmetrically distributed across the tethered bilayer and occupies up to 10% of its volume. The protein CVOs shown in *Figure 7a,b* for stBLMs in contact with protein-containing solutions extend beyond the size of a single PlyCB octamer and could therefore only be modeled by continuous spline-based distributions. In contrast, the CVO profiles in *Figure 7c*, after the buffer rinse, indicate a compact protein conformation within the membrane. With the bilayer structure modeled as a continuous distribution of components (*Shekhar et al., 2011*), we used the PlyCB crystal structure (PDB: 4F87) (*McGowan et al., 2012*) and optimized its distance and orientation at the membrane, as described earlier (*Nanda et al., 2010*). This rigid body modeling approach (*Figure 7c*, black trace) resembled the protein distribution determined in a spline fit (*Figure 7c*, red trace), which suggests that the inserted protein spans the bilayer at a well-defined angle that leads to a match with the bilayer thickness. Data resampling (*Nanda et al., 2010*) determined the most likely model for the protein within the bilayer and established confidence limits for the model parameters. These are indicated as dashed lines in *Figure 7*.

## PlyCB$_{R66}$ is critical for the groove conformation that binds the phospholipid headgroup

The crystal structures of both the PlyC holoenzyme and the PlyCB octamer (PDB: 4F88 and 4F87, respectively; *Figure 3a*) reveal a shallow, charged groove on the PlyCB membrane-binding surface surrounded by residues R29, K59, and R66 (*Figure 3a*, inset). In combination with the residues

**Table 1.** Best-fit parameters from fitting the EIS data in *Figure 4c* to an equivalent circuit model (ECM).

Specific capacitances and resistances were determined after normalization by the geometric electrode surface area ($a$ = 0.32 cm$^2$) and multiplication by the roughness factor ($\beta$ = 1.4).

| ECM parameters | As-prepared bilayer (DOPC/DOPS = 7:3) | Bilayer + PlyCB (480 nM) |
|---|---|---|
| $R_{sol}$ (Ω) | 129.0 ± 0.2 | 106.0 ± 0.3 |
| $C_{stray}$ (nF) | 7.0 ± 0.1 | 13.0 ± 0.3 |
| $C_{mem}$ (µF/cm$^2$) | 0.710 ± 0.002 | 0.739 ± 0.004 |
| $R_{def}$ (kΩ cm$^2$) | 60.20 ± 0.06 | 43.98 ± 0.12 |
| $CPE_{def}$ (µF/cm$^2$) | 8.17 ± 0.06 | 8.97 ± 0.12 |
| $\alpha_{def}$ | 0.925 ± 0.003 | 0.870 ± 0.006 |

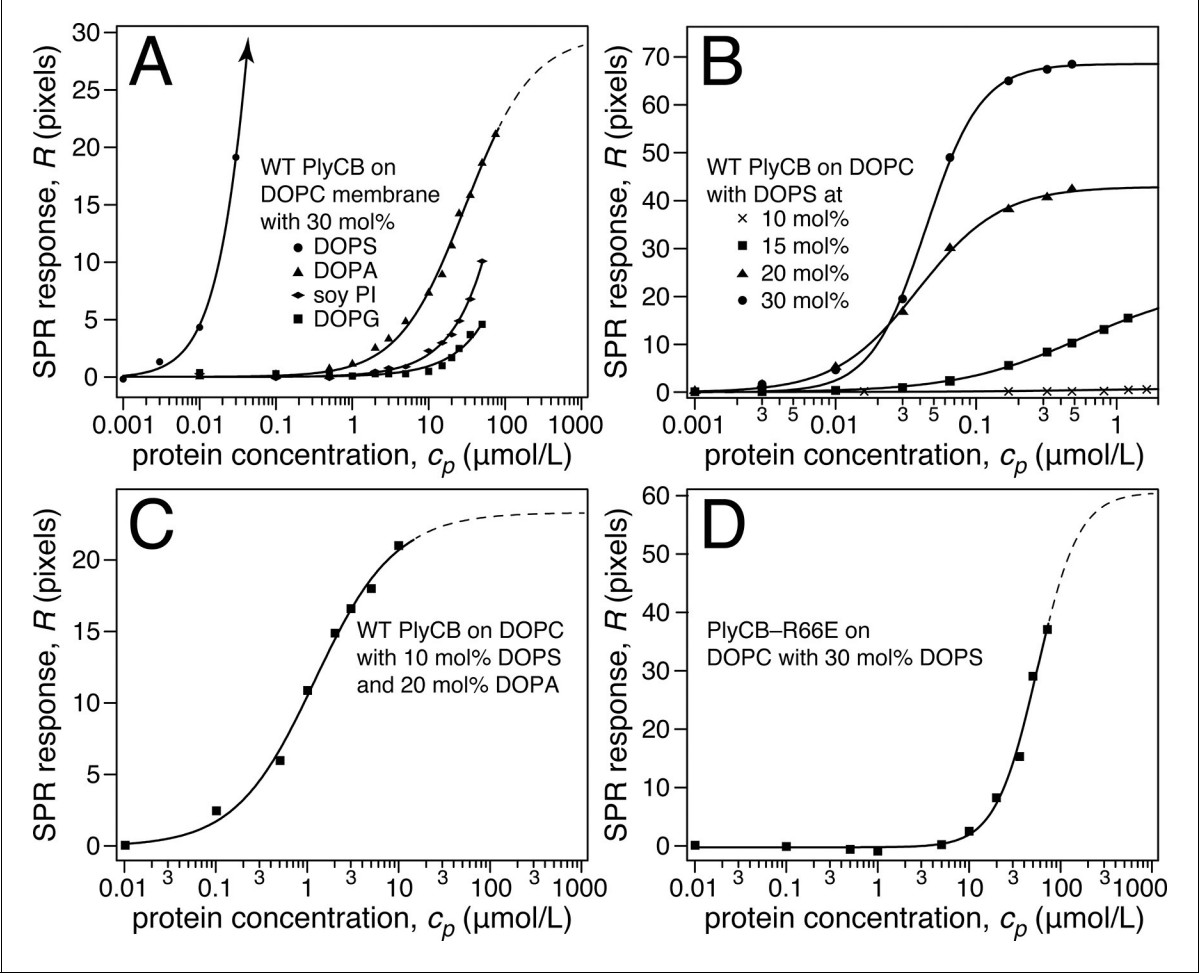

**Figure 5.** Roles of specific interaction of PlyCB with PS and electrostatic interaction in membrane binding. (**a**) Binding isotherms of WT PlyCB to membranes with different anionic phospholipids of the same concentration. The isotherm for DOPC:DOPS = 70:30 is only partially shown. Buffer rinses following protein incubations removed bound PlyCB from PG, PI, and PA-containing membranes as in the experiment shown *Figure 4a*. (**b**) WT PlyCB binding to binary membranes with DOPC and various concentrations of DOPS. (**c**) WT PlyCB binding to a ternary membrane composed of DOPC: DOPA:DOPS = 70:20:10. (**d**) PlyCB_R66E binding to a binary stBLM containing DOPC with 30 mol% DOPS. Data at $c_p > 10$ µM were corrected for protein contributions to the optical index of the buffer (see 'Materials and methods').

identified to affect intracellular translocation, this suggests that the groove is a binding site for anionic lipids. To explore this hypothesis, a 1.7-Å X-ray crystal structure of PlyCB_R66E was determined (PDB: 4ZRZ). Superimposing this structure on that of a high-resolution structure of WT PlyCB (PDB: 4F87) reveals that the mutated E66 side chain maintains the same location as R66 in the WT protein, and a hydrogen bond with E36 is preserved (*Figure 8a*). However, R29 of PlyCB_R66E adopts a new conformation and forms H bonds with E66 and E36, effectively collapsing the groove.

Cocrystallization of PlyCB with PS headgroup moieties and other candidate ligands, and soaking of these into PlyCB crystals, failed to show ordered ligands beyond the putative phosphate site of *Figure 8a*. As an alternative method, we used RosettaLigand (*Meiler and Baker, 2006*) to model PS and PE docking to one subunit of the PlyCB octamer near the groove surrounded by R29, K59, and R66. Because the burial of the lipid fatty tails cannot be adequately accounted for in this approach, these ligands were trimmed to three different lengths (details, see 'Materials and methods'), with the longest construct retaining the α carbons of the fatty acid chains. In all, approximately 30,000 dockings were performed with each ligand. Possible complexes were selected using two criteria. The first candidate set was selected of models with the lowest interface energy scores (Rosetta total energy of the complex minus the sum of the energies of the isolated protein and ligand) and is

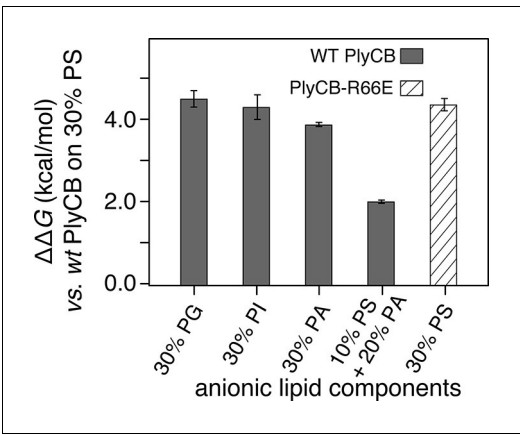

**Figure 6.** PlyCB affinities for anionic lipid components in terms of free energy of membrane binding. Membranes with 30 mol% anionic lipids of various compositions are compared to WT PlyCB on 30 mol% PS. This graph visualizes that WT protein was bound to PG, PI, or PA much more weakly than to PS (positive deviations of $\Delta\Delta G$ indicate lower affinities) and that a low concentration of PS with a larger concentration of PA restored some of the protein binding. PlyCB$_{R66E}$ bound to 30 mol% PS with a similarly low affinity as that of WT PlyCB to 30 mol% PI.

composed of 152 PS and 125 PE candidates. Each of the candidates was visually inspected to exclude models in which (1) the phosphate group binds outside of the potential binding groove or (2) there would be unavoidable clashes if the fatty acid tails were complete, narrowing down the count of viable docking poses to 36 PS and 39 PE complexes. The best accepted PS model had an interface energy 3.74 kcal/mol lower than that of the best model for PE (*Figure 8b*). In 11 PS candidates, including the three best models with different tail constructs, the seryl carboxyl formed a salt bridge with R29 and there was also a salt bridge between the phosphate group and R66. In 18 PE candidates, including the three best models with different tail constructs, there was a salt bridge between the amine and E63, as well as between the phosphate group and R66. These electrostatic interactions dominated the energetics, and resulted in opposite orientations of PE and PS relative to the protein. A summary of the candidate set 1 is given in *Table 3*.

Since energy criteria may not be reliable, the candidate set 2 was compiled by selecting the longest construct models that were excluded from candidate set 1 but contained at least one H bond or salt bridge between the docking site residues and the ligand headgroup or phosphate. This method generated 29 PS and 9 PE candidates. Considering the interfacial H bond energies and their bond lengths, 5 PS and 2 PE models were identified as good candidates that were missed in candidate set 1 (*Table 4*).

**Table 2.** Best-fit parameters from fitting *Equation (1)* to SPR data shown in *Figure 5*.

| Protein | stBLM composition | $K_d$ (μM) | $R_\infty$ (ng/cm$^2$) | $N$ |
|---|---|---|---|---|
| WT PlyCB | DOPC:DOPG 70:30 | >> 50 | n.d. | n.d. |
| | DOPC:soy PI 70:30 | > 50 | n.d. | n.d. |
| | DOPC:DOPA 70:30 | 26.9 ± 2.9 | 170 ± 8 | (*Langmuir fit*) |
| | DOPC:DOPS 90:10 | >> 50 | n.d. | n.d. |
| | DOPC:DOPS 85:15 | 0.49 ± 0.02 | 128.8 ± 2.7 | 1.0 ± 0.1 |
| | DOPC:DOPS 80:20 | 0.04 ± 0.01 | 251.7 ± 0.4 | 1.5 ± 0.1 |
| | DOPC:DOPS 70:30 | 0.04 ± 0.01 | 402.5 ± 0.5 | 2.2 ± 0.2 |
| | DOPC:DOPA:DOPS 70:20:10 | 1.2 ± 0.1 | 134.6 ± 1.2 | (*Langmuir fit*) |
| PlyCB$_{R66E}$ | DOPC:DOPS 70:30 | 62.2 ± 22.2 | 331 ± 72 | 2.0 ± 0.05 |

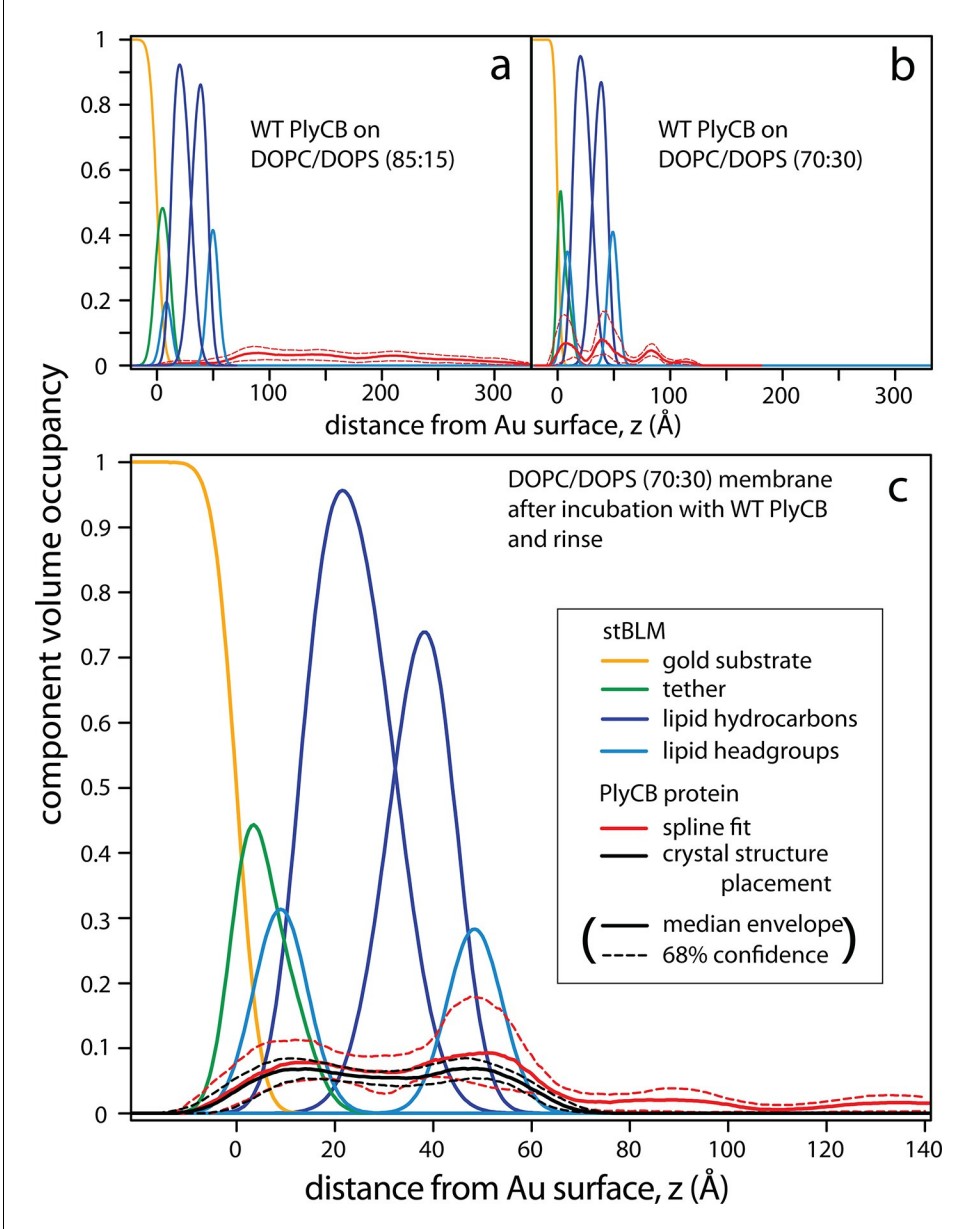

**Figure 7.** PlyCB forms distinct membrane–protein complexes at high and low PS content of the bilayer. CVO profiles (*Heinrich and Lösche, 2014*) of membrane-bound PlyCB on DOPC stBLMs containing (**a**) 15 mol% and (**b, c**) 30 mol% DOPS. Neutron reflection experiments were performed on neat bilayers and on bilayers in contact with protein-containing solutions, and co-refined (protein concentration in panels **a** and **b**: 5 μM and 600 nM, respectively). Panel (**c**) shows the same stBLM as in (**b**) after a buffer rinse to remove loosely associated protein from the membrane surface. Only the structures of the stBLMs in the presence of PlyCB protein are shown. The CVO profile in (**a**) shows accumulation of protein outside the bilayer, but no incorporation into the membrane. In distinction, the protein inserts deeply into the PS-rich bilayer, as indicated by the CVO profiles in panels **b** and **c**. The legend box in (**c**) applies to all panels. Confidence limits of the CVO profiles (not shown for the lipid components but indicated by dashed lines for the protein distributions) were determined by fitting the experimental data with a Monte Carlo Markov Chain.

## Internalization of PlyC depends on caveolae-mediated endocytosis

Since this is the first report of the internalization of an endolysin into epithelial cells, we sought to further explore whether internalization occurs via an active or passive process. Addition of 20 μg/ml

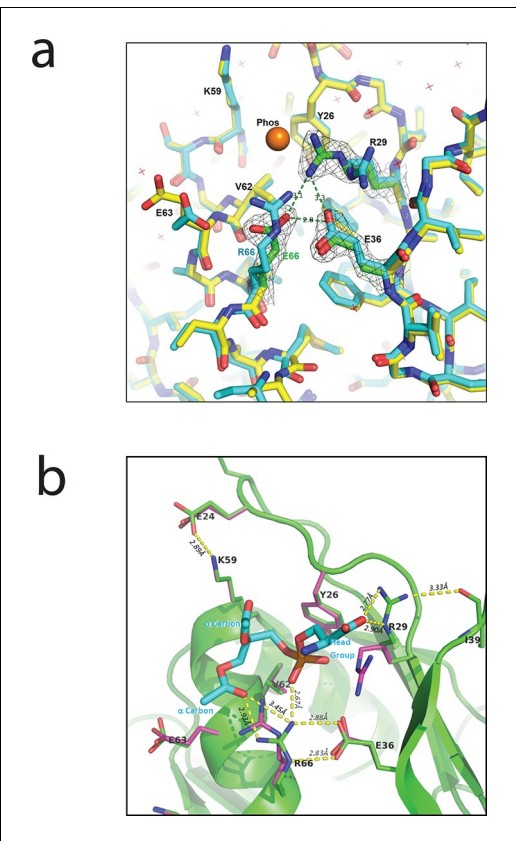

**Figure 8.** Interaction between PlyCB and PS requires structural integrity of a cationic binding groove. (**a**) Superimposed crystal structures of WT PlyCB (PDB: 4F87, cyan carbons) and PlyCB$_{R66E}$ (PDB: 4ZRZ, yellow, with green to emphasize the key residues R29, E36 and the mutated E66) in the groove region. These are both high-resolution structures (1.4 Å and 1.7 Å, respectively) of PlyCB alone. The orange sphere shows the location of a putative phosphate site observed in WT PlyCB in the presence of phosphate or phosphate-bearing ligands. This site holds a water molecule in the WT structure, but an empty site was observed in PlyCB$_{R66E}$, apparently disrupted by the mutation. Mesh shows 2Fo-Fc electron density contoured at 1.6 σ for the three key residues in the mutant structure. The H-bond interaction between R66 and E36 is maintained in the mutant, suggesting that the observed side-chain conformations represent a stable configuration. (**b**) Local structure around the PS docking pose with lowest interface energy, as determined with Rosetta. The PS (cyan carbons) with partially removed fatty acid tails binds to the putative docking site adjacent to R29, K59, and R66. The headgroup and two α carbons of the fatty acid chains are labeled in cyan. The PlyCB backbone is shown in green, with the side chains in the docking site (sticks with green carbons) together with their superimposed conformations from the holoenzyme crystal structure (PDB: 4F88, magenta carbons). Interestingly, the conformation of PlyCB with a double H-bond between R66 and E36, which is the conformation seen in the PlyCB-alone high-resolution structure PDB: 4F87 (*Figure 3a*, *6a*), is reproduced in the PS docking. Proposed H bonds and salt bridges, in the calculated PS docking, as well as the distances between the corresponding atoms are also shown (dashes and labels).

fluorescently labeled PlyCB readily produced punctate intracellular staining in A549 cells within 30 min at 37°C, but not at 4°C (*Figure 9a,b*), indicating energy-dependent, active transport. We also observed that PlyCB colocalizes with cholera toxin subunit B (CTxB), a lipid raft marker that binds GM1 gangliosides (*Latif et al., 2003*), and both CTxB and PlyCB internalization were inhibited by chelation of cellular cholesterol after pre-treatment with filipin III (*Figure 9c,d*). This suggests that PlyCB entry is mediated by a lipid-raft-dependent process such as caveolae-mediated endocytosis (*Lee et al., 2008*). On the other hand, we ruled out other possible internalization pathways. Cytochalasin D, a macropinocytosis inhibitor that induces depolymerization of actin filaments (*Wakatsuki et al., 2001*), did not affect PlyCB internalization, nor did PlyCB colocalize with transferrin, a ubiquitous marker of clathrin-dependent endocytosis (data not shown). Furthermore,

**Table 3.** Summary of the best set 1 candidate models by interface energy.

| Ligand | Set | Best model by interface energy | | H-bond and Salt bridges Interactions | Distances between corresponding atom pairs (Å) |
|---|---|---|---|---|---|
| | | Interface energy score (kcal/mol) | Rosetta total energy (kcal/mol) | | |
| PS | 1A | -6.64 | -1464.65 | Head - R29 | 2.90, 2.77 |
| | | | | P - R66 | 2.67 |
| | | | | Tail - R66 | 3.45, 2.93 |
| | 1B | -3.06 | -1461.49 | Head - R29 | 2.84, 2.90, 3.32 |
| | | | | P - R66 | 2.89, 3.32 |
| | 1C | -3.15 | -1461.72 | Head - R29 | 2.87, 3.43 |
| | | | | Head - E36 | 2.95 |
| | | | | P - R66 | 2.86 |
| PE | 1A | -2.90 | -1465.81 | Head - E63 | 2.92 |
| | | | | P - R66 | 3.19 |
| | | | | Tail - R66 | 2.76, 3.24 |
| | | | | Tail - R29 | 2.92 |
| | 1B | -2.92 | -1464.62 | Head - E63 | 2.95 |
| | | | | P - R66 | 2.88, 3.43 |
| | 1C | -2.61 | -1466.02 | Head - E63 | 2.98 |
| | | | | P - R66 | 2.84, 3.61 |

Ligand in **set 1A** has the longest tail construct (kept up to the α carbon). Ligand in **set 1B** only has head group and phosphate. Ligand in **set 1C** has one carbon after the ester linkage. **Head** is the ligand head group. **P** is the ligand phosphate moiety. **Tail** includes the ligand glycerol moiety and two fatty acid chains.

monodansylcadaverine, an inhibitor of clathrin-dependent endocytosis, did not prevent PlyCB internalization (data not shown).

To determine whether PlyCB utilizes the endocytic pathway for transport within epithelial cells, we analyzed the distribution of labeled protein to different subcellular localizations. Confocal Z-stack analysis (*Figure 9e,f*) showed internalized PlyCB partially colocalized with early endosomal and lysosomal compartments but was also found in vesicles not associated with these compartments. Taken together, these results and *Figure 2c* suggest that the internalized PlyC endolysin utilizes various intracellular transport mechanisms, some leading to lysosomal degradation but others to vesicle fusion with intracellular *Spy*. Further research will be needed to better understand the PlyC mechanism(s) of transport and intracellular interactions with *Spy*.

## Discussion

While extracellular *Spy* remains uniformly sensitive to penicillin, treatment failures are common because *Spy* can invade human host cells and adopt an intracellular niche (*LaPenta et al., 1994*; *Marouni and Sela, 2004*; *Nobbs et al., 2009*) where the antibiotic is unable to penetrate. Internalization is implicated in recurring streptococcal infection because *Spy* can escape from endosomes and enter the cytosol in a streptolysin O-dependent manner, induce apoptosis of host cells, and subsequently repopulate the mucosal surface after antibiotic prophylaxis ends (*Ogawa et al., 2011*; *O'Seaghdha and Wessels, 2013*; *Sharma et al., 2016*). Consequently, there is growing interest in antimicrobial agents that can combat intracellular streptococci.

Bacteriophage-encoded endolysins clearly show therapeutic potential against extracellular bacteria in vitro and in vivo, although to date, there have not been reports of wild type endolysins internalizing a eukaryotic cell. Accordingly, there have been extensive efforts to deliver these enzymes across the plasma membrane, the physical barrier that prevents internalized bacteria from being targeted by antimicrobials. CPPs have been suggested as fusion tags for endolysins to direct them against intracellular pathogens (*Borysowski and Gorski, 2010*). Toward this end, we did not expect

**Table 4.** Good models in set 2 candidates.

| Ligand | Model ID | H-bond and Salt bridges Interactions | Distances between corresponding atom pairs (Å) |
|---|---|---|---|
| PS | gp7_D0145 | Head - R29 | 2.94 |
| | | P - R66 | 2.91 |
| | | Tail - R66 | 2.92 |
| | gp7_D0430 | Head - R29 | 2.79 |
| | | P - R29 | 2.74 |
| | | P - R66 | 2.85, 2.85 |
| | | Tail - K59 | 2.90 |
| | gp8_D0131 | Head - R66 | 2.86, 2.94 |
| | | P - 29 | 2.83 |
| | gp1_F0153 | Head - R29 | 2.83, 2.88 |
| | | P - R66 | 2.93 |
| | gp9_F0363 | Head - R29 | 2.61 |
| | | P - R66 | 2.96, 3.08 |
| | | Tail - R66 | 2.60 |
| PE | gp1_F0391 | Head - E36 | 2.84 |
| | | P - R29 | 2.89 |
| | gp9_F0228 | Head - E36 | 2.95 |
| | | P - R29 | 2.90 |

**Head** is the ligand head group. **P** is the ligand phosphate moiety. **Tail** includes the ligand glycerol moiety and two fatty acid chains.

WT PlyC to internalize cells, and therefore fused PlyC to TAT, one of the best characterized CPPs. In comparing the ability of TAT-labeled PlyC and WT PlyC (used as a control) to lyse intracellular *Spy* (data not shown), we found that *both* lysed the pathogen inside of cells, inspiring the investigations reported here.

PlyC is part of the lytic system of the $C_1$ bacteriophage and, as such, has evolved to degrade the streptococcal peptidoglycan for release of progeny phage. Nonetheless, our results suggest a moonlighting role based on structural motifs within the PlyCB subunit whereby a cationic binding groove is a central determinant of interaction with plasma membranes and subsequent internalization. We propose that PlyC internalization is a three-step process that involves (1) membrane binding through electrostatic interaction and specific ligation with PS, (2) cellular entry by caveolae (or lipid raft)-dependent endocytosis, and (3) cytosolic transport where PlyC comes into contact with intracellular *Spy*. Interestingly, TAT-mediated protein transduction was reported to occur in a similar pathway (**Gump and Dowdy, 2007**), suggesting that PlyC internalization is comparable to cellular uptake of TAT. Whether eventual co-localization between PlyC and *Spy* involves endosomal escape or phagolysosomal fusion is currently unknown and requires further studies.

Our SPR studies show that PS plays an eminent role in PlyCB membrane binding and suggest it is the cellular ligand that mediates protein binding to the plasma membrane. In semi-quantitative terms, this result is echoed in the Rosetta protein-ligand docking models. For PS and PE, only distinguished through the seryl carboxyl, the results for lipids with truncated fatty acid chains suggest that PS has a lower interface energy (by >3.5 kcal/mol) than that of PE. A part of this energetic difference comes from an interaction between the carbonyl group of the fatty acid tail and R66, not observed for PE because of its consistently different orientation relative to the protein. For the shorter fragments (ligand without fatty acid chains and glycerol backbones, see Materials and methods), the best models for PS and PE do not differ significantly in interface energy, at least partly because the tail carbonyl group is absent. Obvious caveats apply to these results: (1) in a real binding event, the phospholipid ligand is membrane-embedded which may restrict viable conformations in the bound state, despite the fact that the membrane is dynamic and flexible, and (2) trimming of the fatty acid tails may eliminate some contribution of the hydrophobic effect in the bound state, which may,

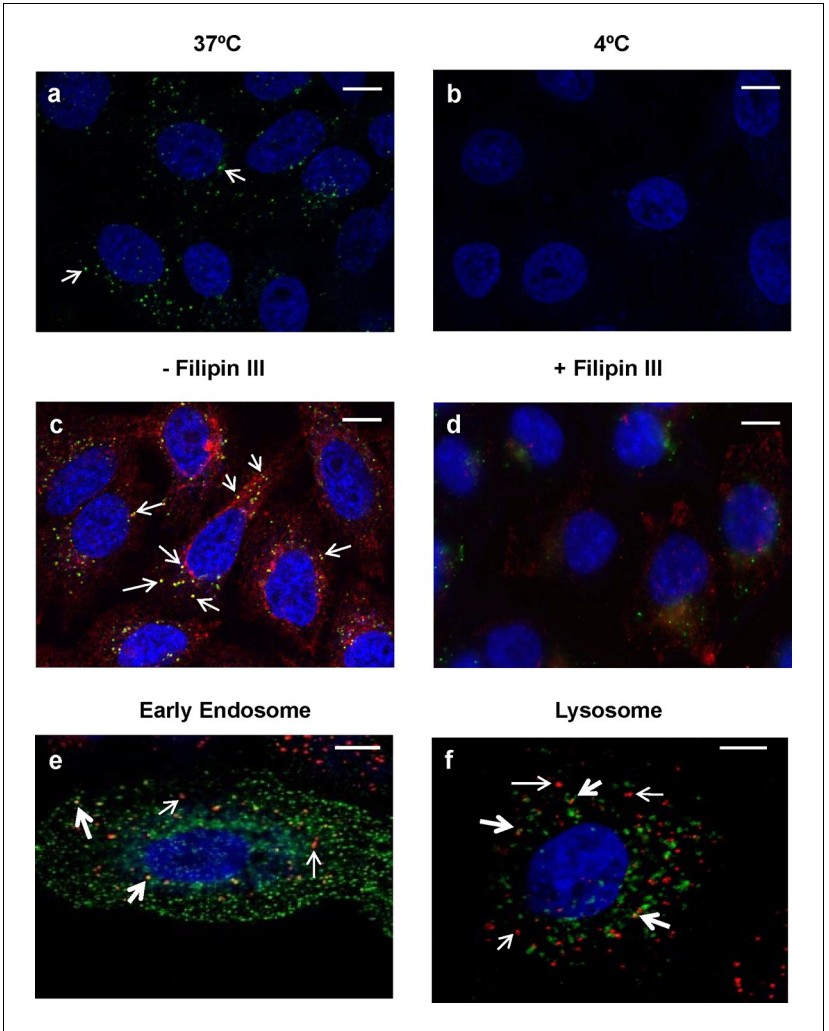

**Figure 9.** Intracellular trafficking of PlyCB depends on caveolae-mediated endocytosis. (**a**) A549 cells were incubated with PlyCB-AlexaFluor488 (green) in serum-free medium F12K for 30 min at 37°C, fixed by 4% PFA in PBS, and stained with DAPI (nucleus, blue). Arrows indicate intracellular PlyCB vesicles. (**b**) PlyCB internalization was not observed when incubating PlyCB-AlexaFluor488 with cells at 4°C. (**c**) Cells were incubated with PlyCB-AlexaFluor488 (green) and CTxB-AlexaFluor555 (red) in serum-free medium F12K for 30 min at 37°C, fixed, and subsequently stained with DAPI (blue). Arrows show co-localization (yellow) of PlyCB and CTxB. (**d**) Cells were pre-treated with Filipin III for 30 min, incubated with PlyCB-AlexaFluor 488 (green) and CTxB-AlexaFluor 555 (red) before being fixed and stained with DAPI. (**e**) A549 cells were transfected with CellLight early endosome-GFP (green) 24 hr prior to treatment of with PlyCB-AlexaFluor555 (red) and fixed after 60 min incubation with PlyCB. PlyCB-AlexaFluor555 co-localizes (yellow) with early endosome compartments as indicated by the bold arrows. The narrow arrows indicate non-endosomal vesicles containing PlyCB. (**f**) Cells were transfected with CellLight lysosome-GFP (green) 24 hr prior to treatment with PlyCB-AlexaFluor555 (red) and fixed after 60 min incubation with PlyCB. The bold arrows points to the co-localization (yellow) of PlyCB with lysosomal compartments. The narrow arrows indicate non-lysosomal vesicles containing PlyCB. Scale bars are 10 μm.

however, be comparable for PS and PE. Despite these caveats, all low-energy candidate models demonstrate strong interactions between the ligand phosphate group and one of the two neighboring arginines – a type of electrostatic interaction that has been experimentally observed in other systems (*Woods and Ferre, 2005*) – in support of the hypothesis that the lipid binding site is adjacent to R29, K59, and R66. We note that the docking calculations are consistent with the experimental PlyCB-membrane binding data, even though the agreement could be fortuitous if various approximations in the docking analysis compensate. Indeed, the free-energy difference between docked PS

and PE compares well with the difference in ΔG observed between PlyCB binding to PS-containing membranes and to membranes without this specific phospholipid ligand. PlyC binding to purely zwitterionic membranes is weak and has not been thoroughly studied, but the binding to non-specific anionic lipids or the binding of PlyCB$_{R66E}$ to PS are good proxies which both show ΔΔG values around 4 kcal/mol (*Figure 6*) – a difference of the same order of magnitude as that observed in the docking result. We therefore suggest that the docking studies, while somewhat simplified, capture the essence of the specificity observed for PS lipid.

PS is an important moiety of eukaryotic cellular membranes, preferentially found on the inner plasma membrane leaflet. Disruption of membrane asymmetry is known to follow apoptosis. On the other hand, recent studies (*Segawa et al., 2011*; *van den Eijnde et al., 2001*) show that membrane inversion is not unique for cell death, and some viruses, such as the vesicular stomatitis virus, utilize PS as the cellular receptor (*Carneiro et al., 2006*). It is therefore plausible that PS can also act as a receptor for PlyC, even more so since PS equilibration across the plasma membrane is more likely for cells that are stressed by pathogen invasion. In an analogous system, annexin A5 binds PS reversibly in a calcium-dependent manner and is often used as a marker for apoptosis. Upon binding to PS-containing membrane patches, annexin A5 forms a convex trimer which nanomechanically elicits membrane invagination and the formation of endocytotic vesicles that are actively transported by the cytoskeleton (*van Genderen et al., 2008*). This translocation pathway is not restricted to apoptotic cells, because other proteins can be trapped in such annexin-A5-mediated portals of entry (*Kenis et al., 2004*). Accordingly, PlyC might penetrate cells in a similar mechanism, as PlyCB possesses eight potential PS binding sites, each on the convex outer surface of the octamer. While further experimental evidence is required, it is tempting to speculate that parallels between the PS-driven membrane interaction of annexins and PlyC may produce membrane invaginations that lead to PlyC endocytosis.

In conclusion, we identify a novel, effective and – as far as currently assessed – safe enzyme-based antimicrobial agent that specifically targets both extracellular and intracellular *Spy*. With the large pool of endolysins that bear distinct functional domains, our results not only show the potential of PlyC as a topical agent to combat refractory streptococcal infection on the skin and mucosal surface, but also provide a rationale to further investigate the intracellular potential of endolysins against other intracellular pathogens in a species-specific manner. Finally, the identification of PlyCB as the shuttle that catalyzes cellular entry of the catalytic PlyCA subunit opens the possibility that distinct payloads can be fused onto the PlyCB scaffold for delivery into mammalian cells.

# Materials and methods

## Bacterial strains and growth conditions

*Streptococcus pyogenes* strain D471, the primary *Spy* strain used in this study, is an M6 strain that has been characterized for its pathogenic mechanisms as well as its ability to be internalized by epithelial cells (*Ryan et al., 2001*). Streptococci were routinely grown in liquid THY medium (Todd-Hewitt broth, supplemented with 1% (wt/vol) yeast extract) or on Columbia blood agar plates (Thermo Scientific, Waltham, MA). For cloning and production of PlyC and its derivatives, a single colony of *Escherichia coli*, strain DH5α or strain BL21(DE3), containing the construct was inoculated and grown in LB-Miller medium (Luria broth containing 10 g/l NaCl) with 100 µg/ml ampicillin. Overnight cultures of bacteria were pelleted and resuspended in fresh LB with 1/3 volume of 80% glycerol before being stored in a –80°C freezer.

## Cloning, generation and purification of mutants

The constructs for pBAD24::*plyC*, pBAD24::*plyCA*, and pBAD24::*plyCB* were previously described (*Nelson et al., 2006*). Mutagenesis studies utilized the QuikChange II Site-Directed Mutagenesis Kit (Agilent Technologies, Santa Clara, CA) and were conducted per manufacturer's instruction. All constructs were verified by DNA sequencing before being transformed into the expression strain BL21 (DE3). The purifications of each construct were performed as previously described (*McGowan et al., 2012*). Mutants were characterized for proper folding by analytical gel filtration as previously described (*Nelson et al., 2006*) and by Fourier Transform Infrared Spectroscopy (FTIR) with an MCT detector in a Hyperion microscope (Nikon, Melville, NY) and a Vertex 80 FTIR (Bruker, Billerica, MA).

## Lytic activity of PlyC and PlyC mutants by spectrophotometric lysis assay

*Spy* strain D471 was grown overnight at 37°C, washed in phosphate-buffered saline (PBS), and resuspended to the desired concentration ($OD_{600 nm}$ = 2.0). One hundred microliter of *Spy* D471 containing suspension was mixed with 100 µl of purified PlyC or mutants (1 µg) in a 96-well plate. The $OD_{600}$ was measured on a SpectraMax 190 spectrophotometer (Molecular Devices, Sunnyvale, CA) every 15 s over a 20 min. period to monitor bacterial lysis. The activity of the PlyC mutants was normalized against wild-type (WT) PlyC endolysin, which represented 100% activity in the 20-min assay.

## Fluorescently labeling of PlyC and mutants

Five microgram of purified PlyC, PlyCA, PlyCB or indicated mutants were reacted with the carboxylic acid, succinimidyl ester of AlexaFluor 555 or AlexaFluor 488 (Invitrogen, Carlsbad, CA) according to the manufacturer's instructions. Unreacted dye was removed from labeled protein by application to a 5-ml HiTrap desalting column (GE Healthcare, Marlborough, MA) equilibrated with PBS. Cross-linked protein samples were subjected to analytical gel filtration on a Superose 12 column (GE Healthcare) calibrated with gel filtration standards (Bio-Rad, Hercules, CA).

## Epithelial cell culture

Human alveolar epithelial A549 cells (Human Lung Carcinoma cell line, CCL-185) were cultured in F-12K medium supplemented with 10% Fetal Bovine Serum (FBS) at 37°C, 5% $CO_2$, and 95% relative humidity. For primary epithelial cell cultures, the experimental protocol received Institutional Review Board approvals from both the Rockefeller University (VAF-0621-1207) and the Weill Cornell Medical College (nos. 0803009695 and 0806009857). Individual patient consent for the use of tissue in research applications was obtained prior to the surgical procedure. Human primary tonsil epithelial cells were isolated and grown in Dulbecco's modified Eagles medium (DMEM) (Thermo Scientific) supplemented with 10% FBS and an antibiotic/antimycotic cocktail (100 µg/ml penicillin, 100 µg/ml streptomycin, 2.5 µg/ml amphotericin B).

## Streptococci/epithelial cell coculture assay

In order to evaluate the intracellular bacteriolytic efficacy of endolysins against internalized *Spy*, we established a coculture assay to quantify *Spy* adherence and invasion. Epithelial cells were grown to 80% confluent monolayers in 24-well tissue culture plates (approximately $2 \times 10^5$ cells/well). An overnight culture of *Spy* strain D471 was washed in sterile PBS, resuspended in serum-free media and the concentration adjusted to $\approx 2 \times 10^7$ CFUs and incubated with epithelial cells at a multiplicity of infection (MOI) of 100 bacterial cells per one epithelial cell for 1 hr. Next, each well was washed 3× with PBS. One hundred microlter of a 0.25% trypsin/0.02% EDTA solution was added to each well to detach cells. Then, 400 µl of a 0.025% Triton X-100 solution in PBS was added to lyse the epithelial cells. Ruptured cells were visible within 5–10 min. Finally, the solution was serially diluted in PBS and plated on Columbia blood agar plates for enumeration of viable CFUs, which represent both adhered and internalized streptococci.

For determination of internalized cell counts, 10 µg/ml penicillin and 200 µg/ml gentamicin were added to the co-culture for 1 hr prior to lysis in order to kill non-adherent and adherent, but not internalized bacteria (as these antibiotics are not taken up by the epithelial cells). Then, internalized bacteria were serially diluted and plated on Columbia blood agar plates for enumeration after epithelial cell lysis. Thus, we differentiated non-adherent, adherent, and internalized *Spy*. To determine the efficacy of endolysins for eliminating internalized *Spy*, post-antibiotic treated co-cultures were washed 3× in PBS and incubated with endolysin for 1 hr before lysis and further enumeration of recovered *Spy* colonies.

## Microscopy

Epithelial cells were seeded onto 12 mm cover slips in 24-well tissue culture plates. When reaching 80% confluence, cells were washed twice with PBS prior to incubation with 20 µg/ml AlexaFluor labeled PlyC, PlyCB, or their mutants in serum-free medium for 30 min. The cells were again washed three times with PBS, fixed by 4% paraformaldehyde (PFA), and mounted with ProLong Gold Antifade Reagent with 4',6-diamidino-2-phenylindole (DAPI) on glass slides for microscopic examination

using an inverted scanning confocal microscope with Argon laser excitation (Carl Zeiss LSM 710, Germany). Images and Z-stacks were obtained with a Plan Apochromat 100×/1.4 objective lens and analyzed with the Zen 2010 digital imaging software (Carl Zeiss). Alternatively, an Eclipse 80i epi-fluorescence microscope (Nikon) with X-Cite 120 illuminator (EXFO, Canada), Retiga 2000R CCD camera, and NIS-Elements software (Nikon) was used for image acquisition and analysis.

## Cellular markers and inhibitors

For the observation of intracellular distribution of internalized PlyCB, A549 epithelial cells were transfected with CellLight early endosome-GFP, CellLight lysosome-GFP, or CellLight actin-GFP (Invitrogen) 24 hr prior to counter staining with 20 µg/ml PlyCB conjugated with AlexaFluor (Molecular Probes) for 30 min. The cells were then washed with PBS twice, fixed with 4% PFA, and mounted with ProLong Gold Antifade Reagent with DAPI on a glass slide for confocal microscopy, as described above.

To determine the internalization pathway, A549 epithelial cells were pre-incubated for 30 min with complete medium in the presence or absence (control) of specific inhibitors of endocytic pathways, including 50 µM monodansylcadaverine (MDC, an inhibitor of clathrin-mediated endocytosis), 1 µg/ml filipin III (an inhibitor of caveolae-mediated endocytosis), or 0.5 µM cytochalasin D (CytD, an inhibitor of micropinocytosis), followed by two PBS washes and incubation with 20 µg/ml PlyCB conjugated with AlexaFluor555 for an additional 30 min. Along with the pharmacological inhibitors, the corresponding fluorescently labeled markers, including transferrin receptor (a marker of clathrin-mediated endocytosis), cholera toxin subunit B (CTxB, a marker of caveolae-mediated endocytosis), and phalloidin (a stress fiber marker of micropinocytosis) were used to confirm the effect of the inhibitors. All inhibitors and markers were from Life Technologies.

## Membrane integrity

For the membrane permeability assay, cells were either permeabilized with 0.02% Triton X-100 (positive control) or incubated with 100 µg/ml PlyC at 37°C for 30 min, washed with PBS twice, fixed with 4% PFA and mounted with ProLong Gold Antifade Reagent with DAPI on a glass slide. For the trypan blue assay, cells were seeded into 24-well tissue culture plates. When 80% confluent, cells were washed twice with PBS prior to incubation with various concentrations of PlyC at 37°C for 30 min. Next, cells were washed 3× in PBS and 100 µl of a 0.25% trypsin/0.02% EDTA solution was added to each well to detach cells from the bottom of wells. The trypsinized cells were then mixed with a 1:1 (vol/vol) trypan blue solution (Thermo Scientific) at 37°C for 30 min, manually counted in a Hausser hemocytometer (Thermo Scientific) and assessed for viability by measuring the proportion of non-viable cells stained by the dye versus unstained viable cells.

## PlyC phospholipid binding prescreen

A phosphoinositide array membrane (PIP Strip, Invitrogen) was used per manufacturer's instructions to qualitatively assess the affinities of His-tagged PlyCB and selected mutants to phospholipids. To detect bound proteins that interact with specific phospholipids, mouse anti-His mAbs followed by goat anti-mouse IgG [HRP] were used and the blot was developed with the LumiSensor Chemiluminescent HRP Substrate Kit (all from GenScript, Piscataway, NJ).

## Lipids and vesicle formation

1,2-dioleoyl-sn-glycero-3-phosphocholine (DOPC), 1,2-dioleoyl-sn-glycero-3-phospho-L-serine (DOPS), 1,2-dioleoyl-sn-glycero-3-phospho-(1'-rac-glycerol) (DOPG), 1,2-dioleoyl-sn-glycero-3-phosphatidic acid (DOPA) and L-α-glycero-3-phosphatidylinositol (soy PI) were from Avanti Polar Lipids (Alabaster, AL). The tether compound, Z 20-(Z- octadec-9-enyloxy)-3,6,9,12,15,18,22-heptaoxatetracont-31-ene-1-thiol (HC18), characterized as described elsewhere (*Budvytyte et al., 2013*), was from Dr. D. Vanderah (Institute for Bioscience and Biotechnology Research, Rockville, MD). Stock solutions of the lipids in chloroform were mixed to obtain targeted lipid compositions. The organic solvent was evaporated by placing the lipid mixtures under vacuum for 12 hr. Dried lipid films were hydrated in a high-salt aqueous buffer (1 M NaCl, 10 mM NaPO$_4$, pH 7.4) to obtain a lipid concentration of ≈5 mg/ml and sonicated until clear suspensions were obtained. These were extruded

through a polycarbonate membrane (Avanti) with a pore size of 100 nm at least 21 times to obtain narrow distributions of unilamellar vesicles.

## Supported bilayer preparation

Sparsely-tethered bilayer membranes (stBLMs) were formed as described earlier (*McGillivray et al., 2007*). Briefly, phosphorous-doped, n-conducting silicon wafers (El-Cat, Ridgefield Park, NJ) for neutron reflection or microscopy glass slides for impedance spectroscopy and SPR measurements were used as substrates. Substrates were cleaned in Nochromix BX10 (Godax Laboratories, Cabin John, MD) solution with concentrated sulfuric acid, followed by extensive rinses with purified water (EMD Millipore, Billerica, NY) and pure ethanol (Pharmco-Aaper, Shelbyville, KY), and dried in a nitrogen flow. They were loaded in a magnetron (ATC Orion; AJA International, North Scituate, MA) and coated with a ≈2 nm Cr adhesion layer and an Au layer with a thickness of ≈45 nm (for EIS or SPR) or ≈15 nm (for NR). Self-assembled monolayers (SAMs) were prepared by overnight incubation of the Au-coated substrates in 0.2 mM (total concentration) ethanolic solution of HC18 and β-mercaptoethanol (βME, Sigma-Aldrich, St. Louis, MO) in a molar ratio of 30:70. Upon removal from the incubation solution, the SAM-covered slides were immediately incubated with a vesicle suspension at high ionic strength for at least 2 hr, followed by a rinse with PBS of low ionic strength (50 mM NaCl) at pH 7.4 that completed stBLM formation by rupture of adhering vesicles under osmotic shock.

## Electrochemical impedance spectroscopy

stBLMs on Si wafers were mounted in an electrochemical cell that implements a three-electrode configuration (*McGillivray et al., 2007*; *Valincius et al., 2006*): the gold-coated substrate, confined by Viton rings to a geometric area of ≈0.32 cm$^2$, served as the working electrode, a saturated silver-silver chloride (Ag|AgCl|NaCl(aq,sat)) microelectrode (M-401F, Microelectrodes, Bedford, NH) was used as a reference electrode, and a 0.25 mm diameter platinum wire (99.9% purity, Sigma-Aldrich) acted as the auxiliary (counter) electrode. The typical roughness factor of the Au film, β ≈ 1.4, was earlier determined (*Budvytyte et al., 2013*). A Solartron 1287A potentiostat and 1260 frequency analyzer were used to measure impedance spectra of the stBLMs. An a.c. voltage with amplitude of 10 mV was applied across the sample while the measurements were carried out within a frequency range of 1 Hz to 100 kHz. Data acquisition was performed with the *Zplot* and *Zview* software packages (Scribner Associates, Southern Pine, NC) and the results analyzed by fitting the data to equivalent circuit models (ECMs) as described (*McGillivray et al., 2007*; *Valincius et al., 2006*). Briefly, the ECM contains elements that represent the solution resistance $R_{sol}$ of the electrolytic buffer and the stray capacitance $C_{stray}$ of the cabling and sample cell. The bilayer is represented by a capacitance $C_{mem}$ parallel to a branch that represents membrane defects by their resistance $R_{def}$ and a constant-phase element (CPE). The CPE has an impedance, $Z = (CPE(i\omega)^\alpha)^{-1}$, that is characterized by its constant-phase element coefficient, $CPE$ (a capacitance per unit area × $t^{\alpha-1}$) and its exponent $\alpha$, which can vary between 0 and 1. Thereby, a CPE becomes a conventional capacitance for $\alpha = 1$. While the membrane capacitance can also be represented by a CPE, we showed in earlier studies that the exponent $\alpha$ is close to one (*McGillivray et al., 2007*). In distinction, $\alpha_{def}$ usually deviates significantly from one. A schematic of the ECM is shown as an inset in *Figure 4c*.

## Surface plasmon resonance (SPR)

A custom-built SPR instrument (SPR Biosystems, Germantown, MD) assembled in the Kretschmann configuration was used to quantify the affinity of PlyCB to stBLMs of various phospholipid compositions. SAM-covered Au-coated glass slides were index-matched with the prism, and stBLMs completed by vesicle fusion in situ. The SPR setup allows for simultaneous EIS measurements, and the quality of the lipid membranes was assessed before each binding experiment. stBLMs with a resistivity below 40 kΩ×cm$^2$ were rejected. In the optical set-up, a fan of monochromatic light (λ = 763.8 nm) hits the sample at a range of incident angles, and a 2D CCD detector records the intensity of the reflected light. The position of the intensity minimum on the CCD defines the resonance angle, recorded as a function of time. The resonance angle corresponding to the neat bilayer is measured as a reference and its change recorded as increasing concentrations of PlyCB in PBS at 150 mM NaCl, pH 7.4, are added to the stBLM. The temperature, typically set to 25°C, is maintained within ± 0.01°C. *SPR Aria* (SPR Biosystems) was used for real-time data recording. The SPR response, *R*, is

measured as a displacement of the reflection minimum on the CCD detector (in pixels). The sensitivity of the instrument has been calibrated with a different protein – the PTEN tumor suppressor (*Shenoy et al., 2012b*) – to be $\Delta\Gamma$ = 5.8 ng/cm$^2$ per pixel of SPR response change. Time courses of $R$ at each protein concentration, $c_p$, were either recorded until equilibrated or fitted to an exponential function for determination of the equilibrium SPR response, $R_{eq}$.

Quantification of low-protein affinities requires high protein concentrations in the buffer. If in excess of ~10 µM protein, this increases the optical index, $n$, of the medium in contact with the sensor surface, thus inflating the readout. Such measurements were corrected as follows. The refractive index increment is d$n$/d$c$ ≈ 0.185 ml/g (*Barer and Joseph, 1954*; *Benesch et al., 2000*; *Shenoy et al. 2012b*). The sensitivity of the instrument is $\Delta n$ = (6.4 ± 0.3)×10$^{-5}$/RU, where the response unit (RU) is the shift of the SPR reflection minimum on the detector in pixels (*Shenoy et al. 2012b*). At a protein concentration, $c_p$ = 50 µM, one thereby expects a change in SPR response of 8.9 ± 0.4 RU due to protein in the buffer. To confirm this calculation, we measured the SPR response of a pure DOPC stBLM to dissolved PlyCB at concentrations up to 75 µM (not shown). Results were consistent with the computed SPR signals under the assumption that the protein does not adsorb to the DOPC bilayer: d$n$/d$c$ = 0.186 ± 0.011 ml/g. SPR at high $c_p$ (> 10 µM) were accordingly corrected.

To quantify the protein affinity to the membranes in terms of its equilibrium binding constant $K_d$ and surface density of bound protein at infinite concentration, $R_\infty$, data were fitted to a Hill function with coefficient $N$,

$$R_{eq} = \frac{c_p^N R_\infty}{c_p^N + K_d} \tag{1}$$

where $N$ = 1 corresponds to the Langmuir adsorption isotherm (*Schasfooort and Tudos, 2008*).

Differences in binding Free Energies of membrane-associated proteins between two SPR measurements that yielded dissociation constants $K_d(1)$ and $K_d(2)$, were determined as

$$\Delta\Delta G = -k_B T \ln(K_d(2)/K_d(1)) \tag{2}$$

## Neutron reflection (NR)

NR measurements were performed at the NGD-Magik reflectometer at the NIST Center for Neutron Research. Reflectivities were typically collected for momentum transfers, $q_z$, between 0.008 and 0.250 Å$^{-1}$. stBLMs were prepared on a SAM-coated silicon wafer and assembled in a flow cell. The flow-through sample cell design allows for in situ buffer exchange on the instrument. Protein-free and protein-loaded membranes were measured in at least two solvent isotopic contrasts that consisted of aqueous buffer prepared with D$_2$O, H$_2$O, or mixtures of the two. For each contrast, acceptable counting statistics are typically obtained after 6 hr. After the measurement of protein-loaded samples, the sample was gently rinsed and the stBLM measurement repeated to characterize protein unbinding. Analysis of NR data was performed using *ga_refl* (*Kienzle et al., 2000-2010*). The structure of the lipid membrane within the stBLM was evaluated in a continuous distribution model (*Shekhar et al., 2011*), and Catmull-Rom splines were used to parametrize the component volume occupancy (CVO) of the lipid components that form the bilayer, as well as the membrane-associated protein (*Heinrich and Lösche, 2014*). A Monte-Carlo resampling technique and Monte-Carlo Markov Chain algorithm (*Kirby et al., 2012*) were used to determine the confidence limits of the fitted parameter values (*Heinrich et al., 2009*).

## Crystal structure

Crystallization of PlyCB$_{R66E}$, diffraction measurement, and structure refinement were as previously described for WT PlyCB (*McGowan et al., 2012*). Briefly, crystals grew in sitting drops at room temperature from purified PlyCB at 14 mg/ml mixed with 43% methylpentanediol, 30 mM ammonium sulfate, 70 mM Na HEPES, pH 6.0. X-rays were produced by a rotating anode generator (MicroMax 007HF, Rigaku, The Woodlands, TX) and diffraction data that yielded a resolution of 1.7 Å were measured by RAXIS IV$^{++}$ image plates and processed using *d\*trek* software (Rigaku). The initial model of the mutant was made by omitting the amino acid side chains 29, 36, 59, 63, and 66 from the precedent structure (PDB code: 4F87). The omitted side chains were imaged using difference

maps and added to the model during refinement. Refinement converged with $R_f$ = 0.25 and rmsd bond lengths of 0.011 Å. The refined mutant structure shown in *Figure 8a* has been deposited in the PDB with accession code 4ZRZ.

## Computational modeling of PlyCB-ligand docking using Rosetta

Initial atomic coordinates of the PlyCB octamer were taken from the PlyC holoenzyme X-ray crystal structure (PDB code: 4F88). To relieve preexisting strain according to the Rosetta's energy function, an energy minimization was performed including all side chains with *RosettaLigand ligand_rpkmin* (*Davis and Baker, 2009*; *Meiler and Baker, 2006*). 216 PS structures were extracted from a fully hydrated DOPC:DOPS (3:1) bilayer MD model (*Shenoy et al., 2012a*). A corresponding set of PE structures was obtained by removing the seryl carboxyl group of PS. Because the burial of the lipid fatty tails cannot be adequately accounted for in this approach, the lipid tails were trimmed as follows: (A) fatty acid tails were kept up to the α carbon. (B) Only the phosphate and headgroup were kept. (C) The lipid headgroup was truncated after the *sn*-3 position of the glycerol. All 216 PS (or PE) conformations in the bilayer model were included in the ligand conformer library. A second conformer library containing 25 to 50 conformations, depending on the construct, was generated using Frog2 web interface (*Miteva etal., 2010*). All docking trajectories were started from a position close to the PlyCB cationic groove containing R29, K59, and R66, featuring the crystallographically observed putative phosphate site shown in *Figure 8a*. Ligand docking poses were sampled within a sphere of 5 Å radius centered at the initial site using *RosettaLigand* (*Davis and Baker, 2009*; *Meiler and Baker, 2006*), which uses a Monte Carlo search algorithm and supports modeling with flexible ligand and protein side chains. For each of the input ligands, 10,000–16,000 models were generated. To compile candidate set 1, the models were ranked by their interface energies. To generate candidate set 2, the models were selected if there existed at least one pair of non-hydrogen atoms between ligand and docking site residues (Y26, R29, E26, K59, V62, E63, and E66) within 3.5 Å distance that were potentially capable of forming an H bond or salt bridge.

## Acknowledgements

This work was supported by the National Institutes of Health (grant number R01 GM110202 to DCN, R01 GM101647 to ML, and R01 AI11822 to VAF) and DARPA to VAF. We are grateful to Silvia Muro for providing inhibitors, Dave Vanderah for providing the HC18 tether lipid, Mariel Escatte, Joseph Kotarek, and Caren Stark for technical support, Georgiy Belov for useful discussions, and Debra Weinstein for editorial assistance. Certain commercial materials, equipment, and instruments are identified in this manuscript in order to specify the experimental procedure as completely as possible. In no case does such identification imply a recommendation or endorsement by the National Institute of Standards and Technology, nor does it imply that the materials, equipment, or instruments identified are necessarily the best available for the purpose.

## Additional information

### Competing interests

VAF, DCN: An inventor of U.S. patents(numbers 6,608,187, 7,582,729, and 7,838,255), which pertain to PlyC. The other authors declare that no competing interests exist.

### Funding

| Funder | Grant reference number | Author |
| --- | --- | --- |
| National Institute of General Medical Sciences | R01GM110202 | Daniel C Nelson |
| National Institute of Allergy and Infectious Diseases | R01AI11822 | Vincent A Fischetti |
| National Institute of General Medical Sciences | R01GM101647 | Mathias Lösche |

The funders had no role in study design, data collection and interpretation, or the decision to submit the work for publication.

## Author contributions
YS, DTG, YY, Conception and design, Acquisition of data, Analysis and interpretation of data, Drafting or revising the article; MB, FH, Conception and design, Acquisition of data, Analysis and interpretation of data; TV, SBL, RDH, DJS, Acquisition of data, Analysis and interpretation of data; DMD, VAF, Conception and design, Drafting or revising the article; JM, Conception and design, Analysis and interpretation of data; ML, DCN, Conception and design, Analysis and interpretation of data, Drafting or revising the article

## Author ORCIDs
Mathias Lösche, http://orcid.org/0000-0001-6666-916X
Daniel C Nelson, http://orcid.org/0000-0003-3248-4831

## Ethics
Human subjects: For primary epithelial cell cultures, the experimental protocol received Institutional Review Board approvals from both the Rockefeller University (VAF-0621-1207) and the Weill Cornell Medical College (nos. 0803009695 and 0806009857). Individual patient consent for the use of tissue in research applications was obtained prior to the surgical procedure.

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
