## [Decision Letter]

Thank you for submitting your work entitled "A Bacteriophage Endolysin that Eliminates Intracellular Streptococci" for consideration by *eLife*. Your article has been favorably evaluated by Richard Losick as Senior editor and three reviewers, one of whom is a member of our Board of Reviewing Editors.

The reviewers have discussed the reviews with one another and the Reviewing Editor has drafted this decision to help you prepare a revised submission.

Summary:

Phage lysins are a novel class of enzymatic activities with several properties that make them attractive for some therapeutic applications. Previously, it was believed, based on dogmatic views of what enters a eukaryotic cell, that such a therapeutic would not be of much value for treating intracellular microbes – often an important reservoir for bacteria escaping from circulating antibiotics. The reviewers agreed that this manuscript describes the important and unexpected finding that one phage lysin appears to enter the eukaryotic cell and there be effective in killing the common human pathogen *Streptococcus pyogenes* residing within. The importance of this is that tonsillar epithelial cells serve as a reservoir for this microbe in asymptomatic carriers and antibiotic treatment failures. There was a consensus of enthusiasm for this manuscript, which both advances knowledge of molecular trafficking into eukaryotic cells, and does so with potentially important practical implications.

Central conclusions:

1) Phage lysin PlyC unexpectedly enters eukaryotic cells where it is active against intracellular *S. pyogenes*.

2) It enters via a process that involves phosphatidylserine (PS).

3) Structural analysis shows the PlyCB to be important for this uptake process, specifically positively charged amino acids on its surface.

Essential revisions:

The following points were raised during review that must be adequately addressed before the paper can be accepted.

1) Physiological relevance of the findings presented in Figure 1. First, what are the levels of *Spy* intracellular colonization achieved in the two models used? How do those compare to levels found in vivo? Does a higher dosage of B30 lead to increased killing of intracellular *Spy*? From comparison with the structures of Ply700 and B30, can one predict the potential of new endolysins to be internalized by eukaryotic cells?

2) It would seem important to test 1h (instead of 30 min, subsection “PlyCB mediates PlyC internalization, leading to colocalization with *Spy* and intracellular lysis of streptococci“, first paragraph) of exposure to 100 µg/ml PlyC, as this is the time of exposure used to assessed the activity of the endolysin on intracellular *Spy* cells. Do increasing exposure time and/or PlyC dose lead to any toxicity toward eukaryotic cells? If yes, what is the potential therapeutic window?

3) Figure 4, as presented, is more confusing than it needs to be. The panels could be larger and re-ordered for more logical presentation. For example, panel D might be better as the last panel because it presents data using a different experimental approach, and panel C could be moved to supplementary material, or presented separately from the SPR data. Lastly, the Figure 4 legend contains a substantial amount of text that is better suited to Results, such as Figure 4 legend text.

4) The manuscript would benefit from editing to remove jargon and to emphasize in the Introduction the importance of intracellular *S. pyogene*s in infection and carriage based published studies.

5) Since primary tonsillar epithelial cells showed a measurably diminished level of internalization and killing efficiency by PlyC compared to cultured A549 cells, could lipid content differences between these cells account for differences in PlyC internalization?

6) Although it is understood that PlyC targets and disrupts the bacterial cell wall and that the PlyCB subunit binds peptidoglycan, there is no mention in this report that *S. pyogenes* membranes also contain phosphatidylserine; might PS be an important receptor in its association with the bacterial surface?

7) In the first paragraph of the subsection “Internalization of PlyC depends on caveolae-mediated endocytosis” and Figure 7. There is some confusion on the statement that filipin III inhibits internalization of PlyCB and CTxB because Figure 7 does show a significant amount of fluorescent-labeled vesicles containing PlyCB. Please clarify.

---

## [Author Response]

Essential revisions: The following points were raised during review that must be adequately addressed before the paper can be accepted. 1) Physiological relevance of the findings presented in Figure 1. First, what are the levels of Spy intracellular colonization achieved in the two models used?

In this manuscript, we report internalization results of *Spy* for two co-culture models. One uses the immortalized A549 lung epithelial cell line and the other one uses primary tonsillar epithelial cells freshly isolated from tonsillectomies. In both co-culture models, ~2% of inoculated *Spy* adhere to the epithelial cells and of those that adhere, 2-10% are internalized, yielding an overall internalization rate of ~0.04 to 0.2%. The adherence/internalization data for the A549 cells is shown in Figure 1—figure supplement 1. Note, while not included in this manuscript, we have done similar co-culture studies with *Spy* on Detroit 562 pharyngeal epithelial cells. In all three co-culture models, the rates of adherence range from 1-5% of the inoculum and rates of internalization range from 1-10% of the adherent streptococci. We did not include the Detroit 562 data in the manuscript because we found the A549 cells easier to grow and used them for all subsequent experiments. Our rates of adherence and internalization in the three models are consistent with the literature as referenced by Ryan et al. and LaPenta et al. Both these manuscripts were already referenced in the original submission, but we now include an expanded statement in our revised manuscript discussing how our adherence/internalization rates compare to previous publications.

Ryan PA, Pancholi V, and Fischetti VA. Group A streptococci bind to mucin and human pharyngeal cells through sialic acid-containing receptors. Infect Immun. 2001. 69(12):7402-12.

LaPenta D, Rubens C, Chi E, and Cleary PP. Group A streptococci efficiently invade human respiratory epithelial cells. Proc Natl Acad Sci U S A. 1994. 91(25):12115-9.

in vivo

Quantification of intracellular *Spy* has only been reported in a few in vivo studies, however, the level is indeed similar to what we and other groups find in co-culture systems. For instance, a murine model of streptococcal skin infection by Kugelberg et al. shows that ~0.01% of inoculated *Spy* persist after Fucidin (a clinically-used antibiotic) treatment, suggesting intracellular *Spy* may have evolved as a mechanism to avoid antibiotic eradication. In addition, in situ image analysis by Thulin et al. reveals that high amount of viable *Spy* present in the soft tissues of human biopsies. Although not ideal, we believe our co-culture model mimics the level of *Spy* colonization/internalization found in vivo.

Kugelberg E, Norström T, Petersen TK, Duvold T, Andersson DI, and Hughes D. Establishment of a superficial skin infection model in mice by using Staphylococcus aureus and *Streptococcus pyogenes*. Antimicrob Agents Chemother. 2005. 49(8):3435–3441.

Thulin P, Johansson L, Low DE, Gan BS, Kotb M, McGeer A, and Norrby-Teglund A. Viable group A streptococci in macrophages during acute soft tissue infection. PLoS Med. 2006. 3(3):e53.

*Does a higher dosage of B30 lead to increased killing of intracellular Spy?*

We have tested B30 up to 100 µg/ml and the results are comparable to those at 50 µg/ml. Hence, only one concentration (50 µg/ml) was selected for depiction in Figure 1.

*From comparison with the structures of Ply700 and B30, can one predict the potential of new endolysins to be internalized by eukaryotic cells?*

The structures for Ply700 and B30 have yet to be solved. However, we do not believe either of these proteins would have internalization potential for a couple of reasons. First, our results demonstrate that PlyCB specifically interacts with phosphatidylserine via a binding groove and no such groove, at least based on primary sequence alignment, is present in Ply700 or B30. Second, even discounting the phosphatidylserine/binding groove results, the electrostatics would make it unlikely for B30 or Ply700 (pI = 4.84 and 4.54, respectively) to interact with the negatively charged lipid membrane. In contrast, PlyCB has a pI of 8.5.

*2) It would seem important to test 1h (instead of 30 min, subsection “PlyCB mediates PlyC internalization, leading to colocalization with Spy and intracellular lysis of streptococci“, first paragraph) of exposure to 100* µg

*/ml PlyC, as this is the time of exposure used to assessed the activity of the endolysin on intracellular Spy cells. Do increasing exposure time and/or PlyC dose lead to any toxicity toward eukaryotic cells? If yes, what is the potential therapeutic window?*

This was an error on our part and has been corrected in the revised manuscript.The exposure time should have said 1h, which was correctly stated in the figure caption, instead of 30 min. In reality, we did these experiments at both 30 min and 60 min, which led to the typo, and found no significant difference in the data between the two time points. Additionally, exposure of PlyC at 100 µg/ml for up to 24 hr did not induce cytotoxicity in our co-culture model as verified by total cell count and the trypan blue dye exclusion assay.

3) Figure 4, as presented, is more confusing than it needs to be. The panels could be larger and re-ordered for more logical presentation. For example, panel D might be better as the last panel because it presents data using a different experimental approach, and panel C could be moved to supplementary material, or presented separately from the SPR data. Lastly, the Figure 4 legend contains a substantial amount of text that is better suited to Results, such as Figure 4 legend text.

The reviewer is correct. Due to the large number of figures/tables alreadyassociated with this manuscript, we made Figure 4 composite figure with many panels. However, the figure/table limitations for *eLife* are more flexible than other journals and as the reviewer points out, this image is more confusing than it needs to be primarily due to the small size of the images. In the revised manuscript, Figure 4 has been broken into three different figures (now new Figure 4, Figure 5, and 6) grouped by similarity in experiments and as the reviewer suggested, the former panel 4D is now the last figure (new Figure 6) in the series. Additionally, as suggested by the reviewer, we have deleted some of text in the figure legends that is already contained in the Results section. The larger size of the image panels, the reordering of the panels, and breaking apart the figure legend into three figure legends should address the review concerns on this point while at the same time making the results more straightforward to read.

*4) The manuscript would benefit from editing to remove jargon and to emphasize in the Introduction the importance of intracellular S. pyogenes in infection and carriage based published studies.*

In the revised manuscript, we have tried to remove all instances of jargon and wehave also added a paragraph and several citations to the Introduction strengthening the importance of intracellular *Spy* to infection and carriage. We thank the reviewer for pointing out these changes that improve the overall quality of the manuscript.

*5) Since primary tonsillar epithelial cells showed a measurably diminished level of internalization and killing efficiency by PlyC compared to cultured A549 cells, could lipid content differences between these cells account for differences in PlyC internalization?*

The reviewer makes an excellent point. Although most published *Spy*adherence/invasion co-culture experiments utilize immortalized epithelial cells including the same A549 cells we investigated, we also sought to include experiments on primary cell lines to account for potential differences that may arise from the immortality of the A549 cells. As stated in response to question 1 above, we saw no significant difference in *Spy* adherence and internalization between primary tonsillar epithelial cells, immortalized A459 lung epithelial cells, and immortalized Detroit 562 pharyngeal epithelial cells (Detroit data not presented in the manuscript). However, as the reviewer points out, we did indeed note a lower level of PlyC killing efficiency in the primary cells compared to the immortalized A549 cells. It is unknown whether these findings are due to experimental variation, differences in growth/metabolism between the primary and immortalized cells, or differences in membrane lipid composition as suggested by the reviewer. Given that our data shows PlyCB binds phosphatidylserine, it is certainly plausible that cell-specific lipid composition may affect PlyC internalization rates and subsequent intracellular killing efficacy. At present, we do not know the lipid composition for these various cell lines, but the revised manuscript now includes a discussion about the differences in PlyC killing efficacy between the various cell lines consistent with the reviewer’s comments.

*6) Although it is understood that PlyC targets and disrupts the bacterial cell wall and that the PlyCB subunit binds peptidoglycan, there is no mention in this report that S. pyogenes membranes also contain phosphatidylserine; might PS be an important receptor in its association with the bacterial surface?*

PlyC binds the streptococcal peptidoglycan via its PlyCB octameric subunit. Whilethe exact nature of the binding epitope on the streptococcal surface is presently unknown, it is strongly believed to be the carbohydrate present within the streptococcal wall teichoic acids, which is consistent with other phage-encoded endolysins that display cell wall binding domains. Although still circumstantial, there is growing evidence that the epitope may be a polyrhamnose moiety within the wall teichoic acids as streptococci sensitive to PlyC all possess the identical polyrhamnose moiety as determined by NMR studies (i.e. *S. pyogenes, S. dysgalactiae*, etc.). Moreover, PlyC readily acts on purified cell walls where no membranes are present and is easily inhibited by cell wall fragments lacking membranes, ruling out an essential role for phosphatidylserine or other membrane lipids. Lysis of *Spy* by PlyC occurs through passive osmotic lysis of the streptococcal membrane in areas where the protective peptidoglycan has been cleaved by PlyC. This “lysis from without” mechanism is shared by all other endolysins derived from phage that infect Gram-positive hosts. After 16 years of studying PlyC, we have yet to find any evidence suggesting binding, interaction, weakening, or direct lysis of the streptococcal membrane by PlyC or PlyCB. Thus, as stated in our Discussion, we believe that PlyCB uses two completely different mechanisms to interact with the *Spy* peptidoglycan and the eukaryotic membrane, the latter being a moonlighting function.

To further the discussion above, we searched the literature for data concerning the lipid composition of *Spy*. Although there are only a few studies that investigate the lipid composition, they suggest the *Spy* membrane is primarily composed of glycolipids and the only identified phospholipids are phosphatidylglycerol and cardiolipin (Rosch et al. and Cohen et al.). We could not find any evidence in the literature suggesting that phosphatidylserine is a component of the *Spy* membrane, further supporting our contention that phosphatidylserine binding by PlyCB is a fortuitous moonlighting function.

Rosch JW, Hsu FF, and Caparon MG. Anionic lipids enriched at the ExPortal of *Streptococcus pyogenes*. J Bacteriol. 2007. 189(3):801-6.

Cohen M and Panos C. Membrane lipid composition of *Streptococcus pyogenes* and derived L form. Biochemistry. 1966. 5(7):2385-92.

7) In the first paragraph of the subsection “Internalization of PlyC depends on caveolae-mediated endocytosis” and Figure 7. There is some confusion on the statement that filipin III inhibits internalization of PlyCB and CTxB because Figure 7 does show a significant amount of fluorescent-labeled vesicles containing PlyCB. Please clarify.

We understand the reviewer’s confusion with these figures (Figure 7 in theoriginal manuscript and Figure 9 in the revised submission). First, we should have used the term “intracellular trafficking” or “intracellular transport” instead of “internalization” to describe this set of experiments because it is the trafficking/transport that they were designed to test. The representative fields were chosen to show the diminished intracellular transport of PlyCB (green) and CTxB (red) after treatment with filipin III. In panel 7C, numerous intracellular vesicle containing both PlyCB and CTxB (yellow, signifying co-localization) are seen. In contrast, Figure 7 (i.e. pre-treated with filipin III) show no co-localization of the green PlyCB and red CTxB, suggesting the co-localization and intracellular transport observed in Figure 7 occurred via a lipid raft mechanism. Nonetheless, as the reviewer points out, some formed vesicles of PlyCB can still be observed in the filipin III treated Figure 7. We expected some PlyCB vesicles to form near the surface, which is what is seen if Figure 7, but the lack of co-localization with CTxB or intracellular transport is the key finding. To help clarify this point, we have added additional arrows to Figure 7 (now 9C) to highlight the yellow co-localization, removed the arrows to the green PlyCB in Figure 7 (now 9D) and we have expanded our discussion and implications of these findings. We do admit that future work will be needed on all intracellular pathways to fine tune our model, but the present data does implicate lipid rafts in the current working model for intracellular transport of PlyCB. We hope the changes to the revised manuscript clarify this point.